# *SkipPredict*: When to Invest in Predictions for Scheduling

**Rana Shahout**
Harvard University

**Michael Mitzenmacher**
Harvard University

## Abstract

Expanding on recent work on scheduling with predicted job sizes, we consider the effect of the cost of predictions in queueing systems, removing the assumption in prior research that predictions are external to the system's resources and/or cost-free. Additionally, we introduce a novel approach to utilizing predictions, *SkipPredict*, designed to address their inherent cost. Rather than uniformly applying predictions to all jobs, we propose a tailored approach that categorizes jobs to improve the effectiveness of prediction on performance. To achieve this, we employ one-bit "cheap predictions" to classify jobs as either short or long. *SkipPredict* prioritizes predicted short jobs over long jobs, and for the long jobs, *SkipPredict* applies a second round of more detailed "expensive predictions" to approximate Shortest Remaining Processing Time for these jobs. Importantly, our analyses take into account the cost of prediction. We derive closed-form formulas that calculate the mean response time of jobs with size predictions accounting for the prediction cost. We examine the effect of this cost for two distinct models in real-world and synthetic datasets. In the external cost model, predictions are generated by external method without impacting job service times but incur a cost. In the server time cost model, predictions themselves require server processing time and are scheduled on the same server as the jobs.

## 1 Introduction

Machine learning research is advancing rapidly, reshaping even traditional algorithms and data structures. This intersection has led to the rise of "algorithms with predictions", also called learning-augmented algorithms, where classical algorithms are optimized by incorporating advice or predictions from machine learning models (or other sources). These learning-augmented algorithms have demonstrated their effectiveness across a range of areas, as shown in the collection of papers (git, [n. d.]) on the subject and as discussed in the surveys (Mitzenmacher and Vassilvitskii, 2020, 2022).

Queueing systems are an example where the learning-augmented algorithm paradigm has been applied for scheduling. Several studies have examined queues with predicted service times rather than exact service times, generally with the goal of minimizing the average time a job spends in the system (Dell'Amico et al., 2015; Dell'Amico, 2019; Mitzenmacher, 2019, 2021; Wierman and Nuyens, 2008), and additionally some recent works also consider scheduling jobs with deadlines (Salman et al., 2023b,a). Minimizing response time is crucial in data centers for reduced latency, in cloud computing for an enhanced user experience and optimized resource utilization, and in real-time telecommunication systems for maintaining reliability and performance through quick task processing.

However, existing works do not adequately model the resources required to obtain such predictions. They often assume that predictions are provided "for free" when a job arrives, which may not be a

38th Conference on Neural Information Processing Systems (NeurIPS 2024).

realistic assumption in the practical evaluation of a system. Incorporating the cost of predictions is essential, as it could be argued that the resources devoted to calculating predictions might be more effectively used to directly process the jobs themselves. This perspective challenges the potential efficiency of integrating predictions into real-world queueing systems. As a result, the following questions arise:

*When does the use of predictions, including their computation, justify their costs?*
*Should all jobs be treated uniformly by computing predictions for each one?*

As a simple example, let us consider a model where predictions of the service time arrive with the job and do not affect the arrival or service times, but do introduce a fixed cost $c$ per job, so the total cost per job is the sum of the mean response time and fixed prediction cost. When we look at Shortest Predicted Remaining Processing Time (SPRPT) policy (Mitzenmacher, 2019), which simply always runs the job with the smallest predicted remaining time, the improvement in the cost over FCFS naturally varies with the prediction cost.

In this paper, we focus on the cost of predictions in settings where two stages of predictions are available. We consider the setting of an M/G/1 queueing system: jobs arrive to a single-server queue, according to a Poisson arrival process with general i.i.d service times. We examine two distinct cost models. In the first model, referred to as the external cost model, predictions are provided by an external server, and they do not affect job service time, but we do factor in a fixed [1] cost for these predictions. The expected overall cost per job in this model is the sum of the job's expected response time within the system and the cost associated with the time for prediction. In the second model, referred to as the server time cost model, predictions themselves require a fixed time from the same server as the jobs to produce, and hence a scheduling policy involves also scheduling the predictions. In this model, the expected overall cost per job is determined by the expected response time. As this model integrates the prediction process within the primary job processing system, it offers a different perspective on the cost implications of predictions as compared to the external model. (In particular, for heavily loaded systems, adding time for jobs to obtain predictions could lead to an overloaded, unstable system.)

As adding a single prediction to an M/G/1 model is relatively straightforward, we consider systems where we have *two stages* of prediction. In the first stage, we may simply predict whether a job is short or long. This type of one-bit prediction (studied in (Mitzenmacher, 2021)) is very natural for machine learning, and in practice may be much simpler and faster to implement. We therefore call these "cheap predictions." In a possible second stage, we predict the service time for a job, which we refer to as "expensive predictions," as in practice we expect them to be substantially more costly. (While we focus on these types of two stages, one could alternatively consider variations where the two stages could yield the same type of prediction; e.g. service time, with the cheap prediction being less accurate but consuming fewer resources.) We introduce a scheduling policy, called *SkipPredict* (*Skip or Predict*), which first categorizes jobs into short and long jobs with the first prediction, prioritizes short jobs over long ones, and restricts additional service time predictions exclusively to long jobs. We analyze the effect of the cost of prediction by considering *SkipPredict* with the previously described external cost model and server cost model. Our approach could be applied to applications where the cost is critical, such as data centers, cloud computing systems, and real-time telecommunication systems; in all these applications, we can reduce the expected time in system, optimizing resources and the user experience while accounting for the overall costs.

We compare *SkipPredict* with three distinct previously studied policies, re-analyzing them with prediction costs in the two proposed models. First, we consider First Come First Served (FCFS), which does not require predictions (and hence incurs no cost from predictions). Second, 1bit (Mitzenmacher, 2021), a policy using only cheap predictions, separates jobs into predicted short and long jobs, and applies FCFS for each category, thereby eliminating the need for a second stage of prediction. Third, Shortest Predicted Remaining Processing Time (SPRPT) performs expensive predictions for all jobs, and no cheap predictions. We find that service time predictions are particularly effective in high-load systems. Our analysis shows that *SkipPredict* potentially outperforms the other policies (FCFS, 1bit, SPRPT) in both cost models using real-world and synthetic datasets, especially when there is a cost gap between cheap and expensive predictions. Additionally, we present another alternative algorithm called *DelayPredict* which avoids cheap predictions by running jobs for a fixed period before executing an expensive prediction. *DelayPredict* initially schedules all jobs FCFS but

---

[1]In Section 4.4 we discuss the case of random costs from a distribution.

limits them to a time $L$. Jobs exceeding the limit $L$ are preempted, and given lower priority, and then they are scheduled by SPRPT. We find *SkipPredict* can also perform better than *DelayPredict* when there is a cost gap between these predictions.

While we focus on scheduling, we believe the approach of using multiple layers of predictions may be useful for other algorithms with predictions problems.

## 2 Related Work

We make extensive use of the SOAP framework (Schedule Ordered by Age-based Priority) (Scully and Harchol-Balter, 2018), which was recently developed to analyze age-based scheduling policies. We use this framework to derive mean response time formulas. We provide a brief background to the framework in Section B.1. For scheduling with predictions, Mitzenmacher (Mitzenmacher, 2019) demonstrated that the analyses of various single-queue job scheduling approaches can be generalized to the context of predicted service times instead of true values. Later, Mitzenmacher (Mitzenmacher, 2021) considered scheduling algorithms with a single bit of predictive advice for whether a job is "long" or "short", based on if its size is above or below a certain threshold. This work shows that even small amounts of possibly inaccurate information can yield significant performance improvements. Scully, Grosof, and Mitzenmacher (Scully et al., 2022) design a scheduling approach for M/G/1 queues that has mean response time within a constant factor of shortest remaining processing time (SRPT) when estimates have multiplicatively bounded error, improving qualitatively over simply using predicted remaining service times. Azar, Leonardi, and Touitou study similar problems in the online setting, without stochastic assumptions, and consider the approximation ratio versus SRPT (Azar et al., 2021, 2022).

Our work has a similar flavor to various "2-stage" problems, such as 2-stage stochastic programming and 2-stage stochastic optimization (e.g., (Grass and Fischer, 2016; Kolbin, 1977; Shmoys and Swamy, 2004; Swamy and Shmoys, 2006)). Here, somewhat differently, our two stages are both predictions of service time at different levels of specificity.

Two recent works have explored incorporating costly predictions into online algorithms. (Benomar and Perchet, 2024) examines how online algorithms can optimally use a limited budget to query for advice. (Drygala et al., 2023) studies the trade-off between the expense of acquiring predictions and the operational benefits they offer. We note that these papers address traditional online problems, such as the ski rental problem. However, the queueing setting introduces unique complexities. Notably, much of the literature in this area focuses on budgeted settings, where the number of predictions is limited. In contrast, our work seeks to optimize overall costs by considering prediction and operational expenses.

## 3 Model

We consider M/G/1 queueing systems with arrival rate $\lambda$. The processing times for each arriving job are independent and drawn based on the cumulative distribution $F(x)$, with an associated density function $f(x)$.

**Scheduling Algorithm: *SkipPredict*** *SkipPredict* initially categorizes jobs based on a binary prediction of either short or long, which we refer to as a cheap prediction. Only jobs predicted as long are further scheduled for a detailed expensive prediction to get the predicted processing time. With *SkipPredict*, jobs that are predicted short have priority over all other jobs. Specifically, predicted short jobs are not preemptible and are scheduled based on First-Come, First-Served (FCFS). Jobs predicted to be long are preemptible and scheduled according to the Shortest Predicted Remaining Processing Time (SPRPT) with predicted size given by the expensive prediction.

We focus on a model where, given a modulated threshold parameter $T$, the cheap predictions are assumed to be independent over jobs, a job of true size $x$ being predicted as short (less than $T$) with probability $p_T(x)$. Similarly, the expensive predictions are assumed to be independent over jobs, and they are given by a density function $g(x, y)$, where $g(x, y)$ is the density corresponding to a job with actual size $x$ and predicted size $y$. Hence $\int_{y=0}^{\infty} g(x, y) dy = f(x)$. We consider two different models, external cost, in which the predictions are provided by an external server, and server cost, in which the predictions are scheduled on the same server as the job.

**Single-Queue, External Cost**  In this model, a job can be described by a triple $(x, b, r)$; we refer to this as a job's type. Here $x$ is the service time for the predictor, $b$ is the output from a binary predictor that determines whether the job is short or long, and for any long job, $r$ is the result of a service-time predictor that provides a real-number prediction of the service time. If a job is predicted to be short, we do not consider $r$, and so we may take $r$ to be null. Again, we refer to $b$ as the cheap prediction and $r$ as the expensive prediction.

In this model, the predictions do not affect the service time of the job, and we treat the overall arrival process, still, as Poisson. Accordingly, *SkipPredict* can be viewed as a two-class priority system: Class 1 is for short jobs, managed by FCFS within the class. Class 2 is for long jobs, according to SPRPT using service time prediction. However, we do associate a cost with predictions which is the mean response time to get the prediction. All jobs obtain a cheap prediction at some constant fixed cost $c_1$, and all long jobs obtain an expensive prediction at some fixed cost $c_2$. Accordingly, we can model the total expected cost for predictions per job in equilibrium as $C = c_1 + c_2 z$, where $z = \int f(x)(1 - p_T(x))dx$ is the expected fraction of jobs requiring the second prediction. In general, both $z$ and $c_1$ will depend on our choice of first layer prediction function, and similarly $c_2$ will depend on the choice of second layer prediction function. Therefore, for some parameterized families of prediction functions, we may wish to optimize our choice of predictors. Specifically, letting $T$ be the expected response time for a job in the system in equilibrium, we might typically score a choice of predictors by the expected overall cost per job, which we model as a function $H(C, T)$, such as the sum of the $C$ and $T$.

**Single-Queue, Server Time Cost**  The server time cost model refers to the setting where predictions are scheduled on the same server as the jobs, so there is a server time cost based on a defined policy. Jobs predicted as short are categorized as non-preemptible while in execution, thereby prioritizing their completion before predicting new jobs. However, jobs predicted as long are further scheduled for a detailed expensive prediction. Thus, cheap predictions outrank expensive predictions and long jobs. Similarly, expensive predictions are prioritized over predicted long jobs.

*SkipPredict* now can be viewed as a four-class priority preemptive system. Class 1 (highest priority) is designated for predicted short jobs, managed by FCFS within the class. Consequently, short jobs are non-preemptible, and are prioritized over predictions for new jobs. This priority is appropriate because even if new jobs are predicted to be short, they will run after the already running short jobs (from FCFS). Class 2 is for (entering) jobs awaiting cheap predictions, which are also handled using FCFS. Class 3 is for jobs awaiting expensive predictions, and also uses FCFS. Finally, Class 4 is reserved for predicted long jobs, served using SPRPT using the service time predictions the jobs obtained when they were Class 3. (Note all jobs enter in Class 2, and the either move to Class 1, or to Class 3 and then Class 4.)

**Definition 1.** *Suppose $\mathbb{E}[T]_{ext}^{PS}$, $\mathbb{E}[T]_{ext}^{PL}$, $\mathbb{E}[T]_{srv}^{PS}$ and $\mathbb{E}[T]_{srv}^{PL}$ are the expected response time for predicted short job and predicted long job in the external cost model and the server cost model in equilibrium respectively. Then, the total cost in the external cost model is*

$$(1 - z) \cdot \mathbb{E}[T]_{ext}^{PS} + z \cdot \mathbb{E}[T]_{ext}^{PL} + C$$

*while the total cost in the server cost model is*

$$(1 - z) \cdot \mathbb{E}[T]_{srv}^{PS} + z \cdot \mathbb{E}[T]_{srv}^{PL}$$

*where $z$ is the expected fraction of jobs requiring the second prediction and $C$ is the expected cost for prediction per job.*

## 4 *SkipPredict* Theorems

For *SkipPredict* with a given $T$ in both models, the expected mean response times for a predicted short job ($\mathbb{E}[T]_{<model>}^{PS}$) and predicted long job of true size $x_J$ and predicted size $r$ ($\mathbb{E}[T(x_J, r)]_{<model>}^{PL}$) are given in Table 1. We provide proofs in Appendix C in Lemmas 1, 2, 3and 4. We get $\mathbb{E}[T]_{<model>}^{PL}$ using Lemma 5 and by plugging $\mathbb{E}[T]_{<model>}^{PS}$ and $\mathbb{E}[T]_{<model>}^{PL}$ into Definition 1, we get the total cost of *SkipPredict* in both models. $c_1$ is the cost of the cheap prediction, $c_2$ is the cost of the expensive prediction and $(r - a)^+ = max(r - a, 0)$. The symbols used in the equations are later expressed and described in Table 2.

**Table 1:** *SkipPredict* equations

| Predicted | External Cost | Server Time Cost |
|---|---|---|
| Short | $\dfrac{\lambda\mathbb{E}[{S'_{<T}}^2]}{2(1-\rho'_{<T})} + \mathbb{E}[S'_{<T}]$ | $\dfrac{\lambda\cdot\left(c_1^2 + 2c_1\mathbb{E}[S'_{<T}] + \mathbb{E}[{S'_{<T}}^2]\right)}{2(1-\rho^{srv}_{PS})} + \mathbb{E}[S'_{<T}]$ |
| Long | $\dfrac{\lambda}{2(1-\rho^{ext}_r)^2}\left(\mathbb{E}[{S'_{<T}}^2] + \mathbb{E}[{S'_{\geq T,r}}^2]\right.$ $\left. + a(r)\right) + \displaystyle\int_0^{x_J}\dfrac{1}{1-\rho^{ext}_{(r-a)^+}}\,da$ | $\dfrac{\lambda}{2(1-\rho^{srv}_r)^2}\cdot\left(\mathbb{E}[{S''_{<T}}^2(c_1)]+\right.$ $(c_1+c_2)^2\cdot Q(T,r) + \mathbb{E}[{S''_{\geq T,r}}^2(c_1+c_2)]$ $\left. + a(r)\right) + \displaystyle\int_0^{x_J}\dfrac{1}{1-\rho^{srv}_{(r-a)^+}}\,da$ |

**Table 2:** Description of symbols used in the equations (defined in Appendix C)

| Symbol | Description |
|---|---|
| $\mathbb{E}[S'_{<T}]$ | Expected service time of predicted short jobs |
| $\mathbb{E}[S'_{\geq T,r}]$ | Expected service time of predicted long jobs with predicted size $\leq r$ |
| $\mathbb{E}[S''_{<T}(c_1)]$ | Expected service time of predicted short jobs including prediction cost $c_1$ |
| $\rho'_{<T}$ | Load due to predicted short jobs |
| $\rho^{ext}_r$ | Load due to predicted short jobs, long jobs with predicted size $\leq r$ |
| $\rho^{srv}_{PS}$ | Load due to predicted short jobs and their cheap prediction cost |
| $\rho^{srv}_r$ | Load due to predicted short, long (with predicted size $\leq r$) and prediction jobs |
| $a(r)$ | Integral related to predicted long jobs with predicted size $> r$ |
| $Q(T,r)$ | Probability of a job being predicted as long with predicted size $> r$ |

## 4.1 Proof intuition of *SkipPredict*

**External Cost Model.** Response time is defined as the sum of the waiting time in the queue and the residence time, which is the time from when the job is first processed until completion. For *SkipPredict* in the external cost model, a predicted short job has to wait behind only previous short jobs. This case is straightforward and follows the mean response time in the system for FCFS, which is given by Equation 23.15 in (Harchol-Balter, 2013). Since there is no preemption in this case, the residence time is simply the service time for the predicted short job, $\mathbb{E}[S'_{<T}]$. A predicted long job of true size $x_J$ and predicted size $r$ has to wait behind previous short jobs, expressed by the term $\mathbb{E}[{S'_{<T}}^2]$, and long jobs with a shorter remaining time, expressed by the terms $\mathbb{E}[{S'_{\geq T,r}}^2]$ and $a(r)$ as well as the final term involving the load. The waiting time is represented by the first term. Since we perform preemption for predicted long jobs, there is a preemption delay that causes a slowdown in the residence time due to the load from jobs predicted as short and jobs predicted as long but with a service time prediction of less than $r$.

**Server Time Cost Model.** For the server cost model, a predicted short job has to wait behind previous short jobs and previous cheap predictions. This addition of the cheap prediction, compared to the external cost model, is expressed by terms involving $c_1$. In this case, there is no preemption, so the residence time is simply the service time for the predicted short job, $\mathbb{E}[S'_{<T}]$.

A predicted long job of true size $x_J$ and predicted size $r$ must wait behind previous short jobs along with their cheap predictions, expressed by the term $\mathbb{E}[S''_{<T}(c_1)]$. It must also wait behind cheap and expensive predictions of long jobs that arrive earlier but with a larger remaining time, expressed by $(c_1+c_2)^2\cdot Q(T,r)$, and long jobs with a shorter remaining time frame along with their cheap and expensive predictions, expressed by $\mathbb{E}[{S''_{\geq T,r}}^2(c_1+c_2)] + a(r)$. The slowdown due to preemption is affected by the load from jobs predicted as short, jobs predicted as long but with a service time prediction less than $r$, and the load from all predictions.

### 4.2  *SkipPredict* Models Comparison

In the server cost model, we observe that the mean response times for both predicted short and long jobs are consistently higher than those in the external cost model, because predictions are scheduled on the same server as the jobs. When we set the costs to zero, so there is no cost to predictions, both models yield identical mean response times. This follows from the definitions of $S''_{\geq T,r}(c_2)$ and $S''_{<T}(c_1)$ (Table 3) because with zero costs, these definitions match with those of $S''_{\geq T,r}$ and $S''_{<T}$. Additionally, setting the threshold $T$ to zero in *SkipPredict* results in SPRPT (Shortest Predicted Remaining Processing Time) scheduling. With this threshold, there are no short jobs (i.e., $\mathbb{E}[S'_{<T}] = 0$), necessitating expensive predictions for all jobs. Thus, the mean response time for the predicted long jobs aligns with the mean response time of SPRPT (Mitzenmacher, 2019) (analyzed for both of the two models in Appendix D.1).

We emphasize that while we have derived equations for total costs for both models, comparing the practical implications of these total costs for the two models is challenging. First, resource allocation differs between the models: in the external cost model, predictions are scheduled on a server separate from the one handling the jobs, whereas in the server cost model, both predictions and jobs are scheduled on the same server. Incorporating a fixed cost into the mean response time for the external cost model does not translate directly to service time. This leads to potential differences in the interpretation of the costs of predictions between the two models.

**Baselines.** As baselines, we compare *SkipPredict* with three distinct policies in the two proposed models. These policies are 1) FCFS[2], a non-size-based policy that does not require predictions; 2) SPRPT, which involves performing expensive predictions for all jobs; and 3) 1bit advice (Mitzenmacher, 2021), which uses only cheap predictions, separating jobs into predicted shorts and predicted longs, and using FCFS as a scheduling policy for each category. These policies, along with *SkipPredict*, can be placed on a spectrum based on their prediction costs. FCFS requires no predictions, while SPRPT requires expensive predictions for each job. The 1bit policy and *SkipPredict* are positioned in the middle of this spectrum, with the 1bit policy incurring lower prediction costs than *SkipPredict*. The question becomes, given prediction costs, what is the most cost-effective policy?

To compare all these policies, we analyze SPRPT and 1bit policies in the external cost model and the server cost model. These policies, initially introduced by Mitzenmacher (Mitzenmacher, 2019, 2021), were analyzed without considering the cost of predictions, so in Appendix D.1 we re-analyze them with prediction costs.

Without looking at the formulas, it is intuitive that under even moderate loads, when the cost of the expensive prediction is low (close to $c_1$), the total cost of *SkipPredict* would be greater than that for SPRPT. Also in this case SPRPT would outperform the 1bit approach, as size-based policies are generally more effective than non-size-based ones when predictions are reasonably accurate. However, when the expensive prediction cost $c_2$ is high, *SkipPredict* or 1bit would be better options than SPRPT. Note *SkipPredict* and 1bit have the same response times for jobs predicted to be short, as both schedule these jobs in the same way. Thus, the efficiency of *SkipPredict* over the 1bit approach depends on the value of using an SPRPT-strategy for the remaining jobs.

***SkipPredict* limitations.** *SkipPredict*'s performance depends on the accuracy of its predictions, as poor prediction quality would diminish its effectiveness in practical applications. Our experiments (Section 5) demonstrate a direct correlation between prediction accuracy and overall performance. Regarding robustness, inaccurate cheap predictions could potentially lead to more long jobs delaying short jobs, highlighting the importance of achieving high prediction accuracy for cheap predictions.

### 4.3  What if the cheap predictions are not really cheap?

Another limitation of *SkipPredict* arises in situations where sufficiently low-cost predictions are not available or are not substantially less costly than expensive predictions. In such scenarios, *SkipPredict* may be less effective. However, on the positive side, our work could provide insights into the necessary cost thresholds for achieving performance gains. Here, we also suggest an alternative algorithm, *DelayPredict*, as a potential solution in these cases. Rather than apply predictions to all jobs, *DelayPredict* does not use cheap predictions, while still avoiding expensive predictions for short jobs. *DelayPredict* schedules jobs initially with FCFS, but limits each job to a given limit $L$

---

[2]We could also consider any non-size-based policy such as LCFS or FB.

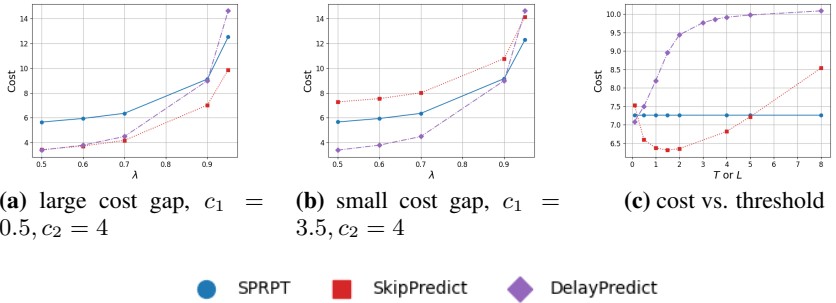

**Figure 1:** *DelayPredict* vs. *SkipPredict* and SPRPT in the external cost model with exponential service times and exponential predictor as described in Section 5 (a) cost vs. arrival rate with $c_1 = 0.5, c_2 = 4, T = L = 1$ (b) cost vs. arrival rate with $c_1 = 3.5, c_2 = 4, T = L = 1$ (c) cost vs. threshold ($T$ for *SkipPredict* and $L$ for *DelayPredict*), $c_1 = 0.5, c_2 = 2, \lambda = 0.9$.

of service time, at which point it is preempted, and treated as a *long job*. At that point, the job can go through expensive prediction and be scheduled based on SPRPT scheduling. A job that finishes before $L$ units of service would, in this setting, be a short job that finishes without any prediction. We define $z' = \int_{x=L}^{\infty} f(x)dx$ as the expected fraction of jobs requiring the expensive prediction and use it in Definitions 1 to analyze *DelayPredict* cost in the external cost model and in the server cost model. We provide mean response times for *DelayPredict* with proofs in Appendix E.

***DelayPredict* vs. *SkipPredict*.** The main difference between *DelayPredict* and *SkipPredict* is in the waiting time. In *SkipPredict*, a predicted short job sees only short jobs while a short job in *DelayPredict* sees all jobs in the queue ahead of it, limited to size $L$. Similarly, in *DelayPredict* a long job has to wait behind incoming jobs (including all other long jobs) for at lest time $L$. Thus, while *DelayPredict* saves the cheap predictions, its waiting time can be higher than *SkipPredict*. Figure 1 shows a setting where there is a cost gap between prediction costs, and *SkipPredict* outperforms *DelayPredict*. However, *DelayPredict* still performs better than SPRPT (with $c_1 = 0.5, c_2 = 4$). When the costs are close, *DelayPredict* is better than both *SkipPredict* and SPRPT, as in this case *SkipPredict* is less effective. Since the waiting time of *DelayPredict* depends on $L$, we see in Figure 1(c) that as $L$ increases, the cost of *DelayPredict* gets higher. (Note for comparison purposes the $L$ for *DelayPredict* and the $T$ for *SkipPredict* are chosen to be the same value.)

## 4.4 Generalization to Non-Fixed Costs

While we assume that the prediction costs as fixed, our approach naturally generalizes theoretically to random costs from a distribution, where that distribution may also depend on the service time of the job. Here we outline the necessary modifications in the analysis for this generalization.

We may consider prediction costs that are assumed to be independent over jobs. The cheap prediction cost is given by a density function $k_1(x, c_1)$, where $k_1(x, c_1)$ is the density corresponding to a job with actual size $x$ and cost of cheap prediction $c_1$. Hence, $\int_{c=0}^{\infty} k_1(x, c)dc = f(x)$. Similarly, the expensive prediction cost is given by a density function $k_2(x, c_2)$, where $k_1(x, c_2)$ is the density corresponding to a job with actual size $x$ and the cost of expensive prediction is $c_2$.

In our analysis of the server cost model with fixed costs, for jobs with rank 4 in the first dimension, we used $r - a$ to encode the second dimension of the rank rather than the predicted remaining service time, which we noted is technically $r - (a - c_1 - c_2)$. When $c_1$ and $c_2$ are fixed, doing so does not change the rankings of jobs, but for non-fixed costs, we would want to use the actual predicted remaining service time $r - (a - c_1 - c_2)$ for the rank function.

# 5 Experimental Results

To gain more insight into when to invest in prediction and how *SkipPredict* compares to other policies, in this section we compare *SkipPredict*, SPRPT, 1bit and FCFS using simulation with real-world and synthetic traces in the setting of the single queue with Poisson arrivals. For real-world traces, we used three traces from Amvrosiadis et al. (Amvrosiadis et al., 2018): Twosigma, Google, and Trinity. Their system leverages machine learning to predict job runtimes in large clusters, utilizing features like user IDs, job names, and input sizes, Appendix F contains more details about the datasets and the predictor. For the synthetic traces, we considered two job service time distributions: exponentially distributed with mean 1 ($f(x) = e^{-x}$) and the Weibull distribution with cumulative distribution $F = 1 - e^{-\sqrt{2x}}$ (which also has mean 1). Also, for the synthetic traces, we use two prediction models where each of the two-stage predictors could be from a different model: 1) Exponential predictions (Mitzenmacher, 2019), where a prediction for a job with service time $x$ is itself exponentially distributed with mean $x$. 2) Uniform predictions (Mitzenmacher, 2019), where a prediction for a job with service time $x$ is uniformly distributed between $(1 - \alpha)x$ and $(1 + \alpha)x$ for a parameter $\alpha$. For the synthetic datasets, in this section we present results for the exponentially distributed service time with uniform predictor. Appendix F contains results for the rest of the combinations.

To handle two-stage predictions, both for the real-world traces and for the synthetic ones, the cheap predictor here returns a single bit by comparing the predicted value with the threshold; this is not how an actual prediction would work, but it is just a test model for simulation. We note that, as part of verifying our equations and our system, we have checked simulation results for single queues against the equations using the "perfect predictor," as the integrals are simpler in this case.

**Cost vs. arrival rate $\lambda$.** Figures 2, 3(b), 3(d), 4(b), 4(d) show the cost using real-world and synthetic datasets. These Figures show that investing in service time predictions is more beneficial at higher arrival rates. *SkipPredict* yields the lowest cost above $\lambda = 0.7$ in all datasets. We should note, however, that under extremely high load in the server cost model, prediction-based scheduling (SPRPT, *SkipPredict* or 1bit) risks overflowing the system (since the average time per job with predictions is larger than 1), making FCFS a better option.

**Cost vs. $T$.** In Figures 2(g), 2(h), 3(a), 3(c) 4(a), 4(c), we compare the cost vs. $T$. As $T$ increases, both *SkipPredict* and 1bit demonstrate reduced costs. However, past a certain $T$ threshold, which depends on the dataset, arrival rate, and actual costs, we observe a rise in costs due to the decreasing number of jobs requiring expensive prediction. This leads to a reduced load for expensive predictions, making job size prediction for scheduling between predicted long jobs less effective. In the synthetic datasets, as $T$ gets large, *SkipPredict* and 1bit become the same, as both serve predicted short jobs similarly, and for large $T$ nearly all jobs are predicted short.

**Accuracy/Cost tradeoff.** To evaluate a setting where one has to choose accuracy levels by accuracy/cost tradeoff, we conducted experiments where we controlled the accuracy, and consequently, the cost, so higher accuracy corresponded to higher prediction costs. Again, the point here is to show a possible use case, where a user might decide from a set of possible available accuracies and costs to optimize the system. The measured total cost is presented a heat map matrix in Figures 5(a), 5(b) using exponential service time and $T = 1$. In this matrix, rows represent the probability of accurate prediction and associated costs for the cheaper model, columns represent the $1 - \alpha$ values of the uniform predictor, and each cell displays the corresponding total cost for that setting. These figures indicate that reducing the accuracy of the cheap prediction is more crucial since the cheap model's prediction is performed for every job. For expensive predictions, the relationship between $\alpha$ values and the total cost is not strictly linear and the cost can be optimized based on expensive prediction costs and number of predicted long jobs.

***SkipPredict* benefit increases with larger cost gaps.** We have found that *SkipPredict* is more cost-effective than other policies when there is a difference between the costs of the two predictions, with a greater gap leading to higher improvement. In Figures 5(c) 5(d), we have exponential service times, $T = 1$, $c_1$ is fixed to the default value, and we change $c_2$ by varying $k$ where $c_2 = kc_1$. For similar or very close costs ($k = 1$), *SkipPredict* is less useful, and SPRPT may be a better option. However, as $c_2$ values increase, *SkipPredict* becomes more cost-effective. FCFS and 1bit are not affected as they do not require expensive predictions.

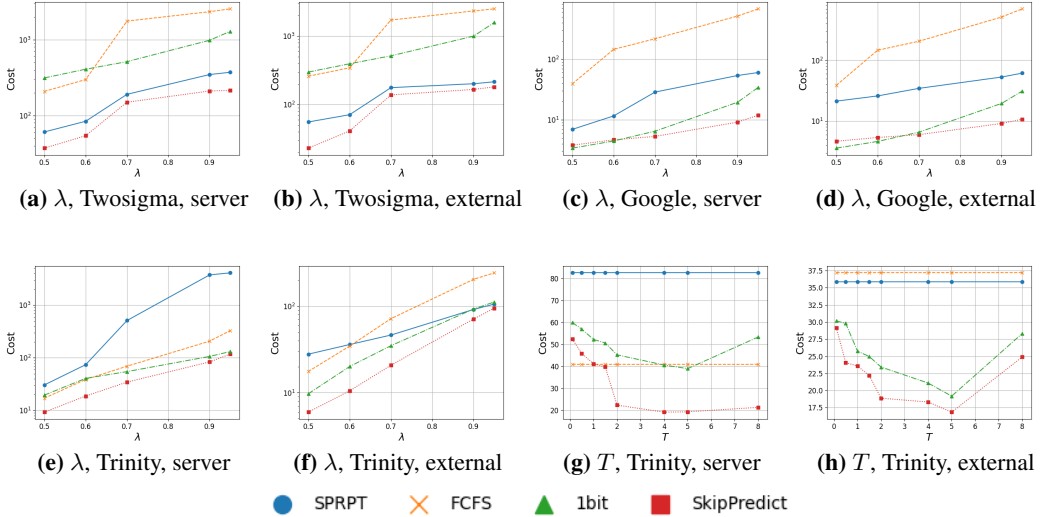

**(a)** $\lambda$, Twosigma, server  **(b)** $\lambda$, Twosigma, external  **(c)** $\lambda$, Google, server  **(d)** $\lambda$, Google, external

**(e)** $\lambda$, Trinity, server  **(f)** $\lambda$, Trinity, external  **(g)** $T$, Trinity, server  **(h)** $T$, Trinity, external

●  SPRPT  ✕  FCFS  ▲  1bit  ■  SkipPredict

**Figure 2:** Cost in the server time model and in the external model using real-world datasets (a-f) vs. $\lambda$ (with $T = 4$) (g-h) vs. $T$ with Trinity dataset (with $\lambda = 0.6$). The default costs for the external model are $c_1 = 0.5, c_2 = 20$, and in the server time are $c_1 = 0.05, c_2 = 0.5$.

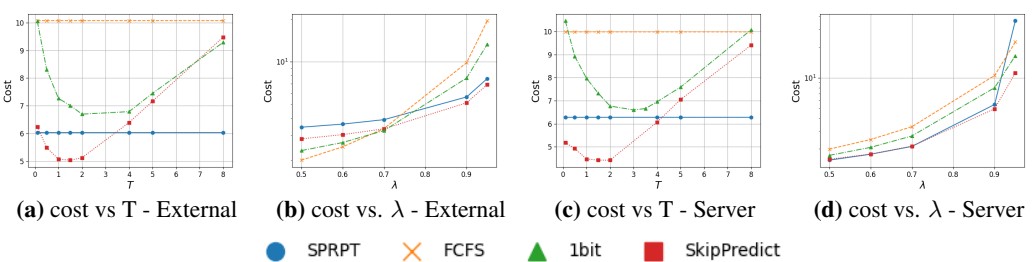

**(a)** cost vs T - External  **(b)** cost vs. $\lambda$ - External  **(c)** cost vs T - Server  **(d)** cost vs. $\lambda$ - Server

●  SPRPT  ✕  FCFS  ▲  1bit  ■  SkipPredict

**Figure 3:** Cost in the external cost and server cost models using uniform predictor, cheap predictor is configured with $\alpha = 0.8$, expensive predictor with $\alpha = 0.2$, service times are exponentially distributed with mean 1. The default costs for the external model are $c_1 = 0.5, c_2 = 2$ and for the server cost model are $c_1 = 0.01, c_2 = 0.05$ (a + c) Cost vs. $T$ when $\lambda = 0.9$ (b + d) Cost vs. $\lambda$ when $T = 1$.

## 6  Conclusion

We have presented *SkipPredict*, the first prediction-based scheduling policy we are aware of that takes into account the cost of prediction. *SkipPredict* is designed for systems where two levels of prediction are available, good (and cheap) and better (but expensive) prediction. While here we have focused on having a binary prediction (short/long) and a prediction of the service time, our framework would also work for other settings. For example, both the cheap and expensive predictions could be for the service time, with the expensive prediction being a more refined, time-consuming variation of the cheaper process (that even takes the cheap prediction as an input). We considered the cost of prediction in scheduling in two models; the external cost model with externally generated predictions, and the server time cost model where predictions require server time and are scheduled alongside jobs. We derived the response time of *SkipPredict* and analyzed total cost formulae in the two cost models for *SkipPredict*. We similarly derived formulae in these models for previously proposed prediction policies where previous analyses ignored prediction costs, namely 1bit and SPRPT, as well as a new policy, *DelayPredict*. We have demonstrated, using both real-world and synthetic datasets, that *SkipPredict* potentially outperforms FCFS, 1bit, SPRPT, and *DelayPredict* in both

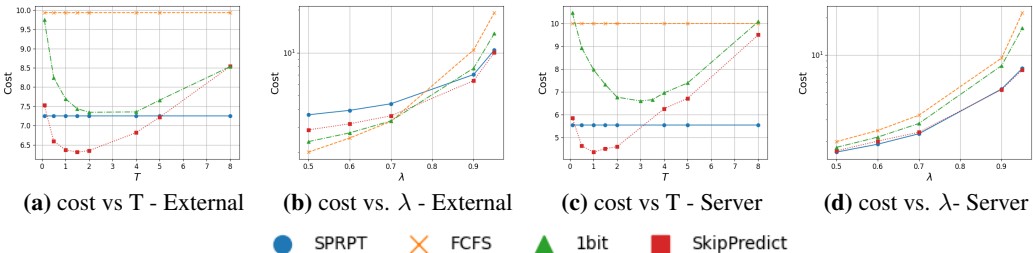

**Figure 4:** Cost in the external cost and server cost models using exponential predictor for both cheap and expensive predictors, service times are distributed exponentially with mean $1$. The default costs for the external model are $c_1 = 0.5, c_2 = 2$ and for the server cost model are $c_1 = 0.01, c_2 = 0.05$.

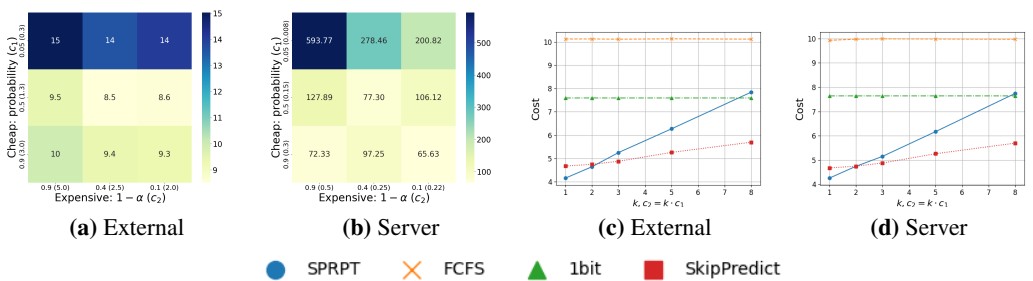

**Figure 5:** (a+b) cost in external and service time models in setting where one has to choose accuracy levels by accuracy/cost tradeoff. (c+d) Cost vs. $c_2$ with $c_1 = 0.5$. We use exponential service time with, $\lambda = 0.9$ and $T = 1$.

cost models, especially when there is a significant cost gap between cheap and expensive predictions. While we focused on analyzing *SkipPredict* for cases with prediction costs, there are many potential variations to explore. These include using separate servers for prediction, offering more than two priority classes, selectively predicting only some jobs with probability to reduce costs while prioritizing between predicted short and long jobs, as well as employing load-based predictions to make predictions only when the queue length exceeds a certain threshold.

While scheduling itself is a foundational problem that provides sufficient motivation for our work, we believe the issue of accounting for the costs of predictions in the "algorithms with predictions" framework (and arguably other similar problems) is understudied. Our idea of choosing one or more predictions, and optimizing the overall cost including that choice, is an approach that we believe will be useful for other similar problems.

**Impact Statement.** This paper presents work whose goal is to advance the field of Machine Learning. There are many potential societal consequences of our work, none which we feel must be specifically highlighted here.

# Acknowledgments

Rana Shahout was supported in part by Schmidt Futures Initiative and Zuckerman Institute. Michael Mitzenmacher was supported in part by NSF grants CCF-2101140, CNS-2107078, and DMS-2023528.

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

# Appendix

## Table of Contents

## A   Supplementary Figures

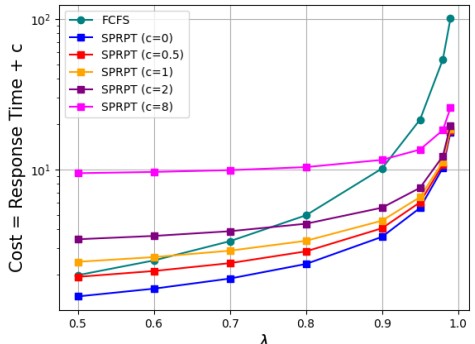

**Figure 6:** Considering prediction cost in SPRPT algorithm in M/M/1 system. The cost is the sum of mean response time and fixed prediction cost.

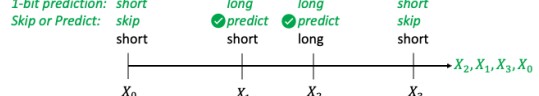

**Figure 7:** The *SkipPredict* algorithm.

# B  Formulas via SOAP Analysis

We employ the SOAP framework (Scully and Harchol-Balter, 2018), a (relatively) recently developed analysis method, to obtain precise formulas for mean response time[3] of *SkipPredict* in both the external cost model and in the server cost mode. While we could analyze the external cost model without SOAP as a two-class system, we choose to use SOAP for a consistent analysis.

## B.1  SOAP Background

The SOAP framework can be used to analyze scheduling policies for M/G/1 queues that can be expressed in terms of rank functions. Recent research by Scully and Harchol-Balter (Scully and Harchol-Balter, 2018) has classified many scheduling policies as SOAP policies. These policies determine job scheduling through a rank, always serving the job with the lowest rank. (In cases where multiple jobs share the lowest rank, the tie is resolved using First Come First Served.) The rank function determines the rank of each job, and it can depend on certain static characteristics of the job, often referred to as the job's type or descriptor. For example, the descriptor could represent the job's class (if the model has different classes), and other static attributes, such as its size (service time). The rank can also depend on the amount of time the job has been served, often referred to as the job's age. A key assumption underlying SOAP policies is that a job's priority depends only on its own characteristics and its age, an aspect that aligns with our model and scheduling algorithm *SkipPredict*. We refer the interested reader to (Scully and Harchol-Balter, 2018) for more details.

SOAP analysis uses the tagged-job technique. That is, we consider a tagged job, denoted by $J$, of size $x_J$ and with descriptor $d_J$. We use $a_J$ to denote the amount of time $J$ has received service. The mean response time of $J$ is given by the sum of its waiting time (the time from when it enters to when it is first served) and the residence time (time from first service to completion). To calculate the waiting time, SOAP considers the delays caused by other jobs, including "old" jobs that arrived before $J$ and "new" jobs that arrived after $J$. A key concept in SOAP analysis is the worst future rank of a job, as ranks may change non-monotonically. The worst future rank of a job with descriptor $d_J$ and age $a_J$ is denoted by $rank_{d_J}^{\text{worst}}(a_J)$. When $a_J = 0$, the rank function is denoted by $r_{worst} = rank_{d_J}^{\text{worst}}(0)$. Ranks are written in ⟨*angle brackets*⟩.

In the SOAP framework, waiting time is shown to be equivalent to the transformed busy period in an M/G/1 system with arrival rate $\lambda$ and job size $X^{\text{new}}[rank_{d_J}^{\text{worst}}(a)]$ [4]. The initial work of this period represents the delay caused by old jobs. To deal with the delay due to old jobs, SOAP introduced a transformed system where jobs are categorized based on their rank. *Discarded* old jobs, exceeding the rank threshold $r_{worst}$, are excluded from the transformed system. *Original* old jobs, with a rank at or below $r_{worst}$, are considered as arrivals with rate $\lambda$ and a specific size distribution $X_0^{\text{old}}[r_{worst}]$ [5]. *Recycled old jobs*, currently at or below $r_{worst}$ but previously above this threshold, are treated as server vacations of length $X_i^{\text{old}}[r_{worst}]$[6] for $i \geq 1$ in the transformed system. As explained later, in *SkipPredict* jobs could only be recycled once, so we only have $X_1^{\text{old}}[r_{worst}]$.

SOAP shows that, because Poisson arrivals see time averages, the stationary delay due to old jobs has the same distribution as queueing time in the transformed M/G/1/FCFS system. This system is characterized by 'sparse' server vacations, where (*original*) jobs arrive at rate $\lambda$ and follow the size distribution $X_0^{\text{old}}[r_{worst}]$.

---

[3]We note that SOAP provides for finding the Laplace-Stieltjes transform of the response time distribution; we focus on the mean response time throughout this paper for convenience in comparisons.

[4]$X^{\text{new}}[rank_{d_J}^{\text{worst}}(a)]$ is a random variable representing how long a new job that just arrived to the system is served until it completes or surpasses $rank_{d_J}^{\text{worst}}(a)$

[5]$X_0^{\text{old}}[r_{worst}]$ is a random variable representing how long a job is served while it is considered *original* with respect to the rank $r_{worst}$

[6]$X_i^{\text{old}}[r_{worst}]$ is a random variable representing how long a job is served while it is considered *recycled* for the $i$ time with respect to the rank $r_{worst}$

**Theorem 1** (Theorem 5.5 of (Scully and Harchol-Balter, 2018)). *Under any SOAP policy, the mean response time of jobs with descriptor $d$ and size $x_J$ is:*

$$\mathbb{E}[T(x_J, d)] = \frac{\lambda \cdot \sum_{i=0}^{\infty} \mathbb{E}[X_i^{old}[r_{worst}]^2]}{2(1 - \lambda \mathbb{E}[X_0^{old}[r_{worst}]])(1 - \lambda \mathbb{E}[X^{new}[r_{worst}]])}$$
$$+ \int_0^{x_J} \frac{1}{1 - \lambda \mathbb{E}[X^{new}[rank_{d_J}^{worst}(a)]]} da.$$

## B.2 Rank functions of *SkipPredict*

The relevant attributes to *SkipPredict* are the size, the 1-bit prediction, and the predicted service time. We can model the system using descriptor $\mathcal{D} = $ [size, predicted short/long, predicted time] $= [x, b, r]$. *SkipPredict* in the external cost model results in the following rank function:

$$rank_{ext}([x, b, r], a) = \begin{cases} \langle 1, -a \rangle & \text{if } b = 1, \\ \langle 2, r - a \rangle & \text{if } b = 0. \end{cases} \tag{1}$$

which uses the first dimension to encode the class priority (short or long), and the second dimension to enforce the priority for each class (FCFS for short jobs, SPRPT for long jobs). In such nested rank function, the first dimension serves as the primary rank, with the priority ordering following a lexicographic ordering.

In the server cost model, *SkipPredict* results in the following rank function:

$$rank_{srv}([x, b, r], a) = \begin{cases} \langle 2, -a \rangle & \text{if } 0 \leq a \leq c_1 \text{ (initial rank, and cheap prediction)}, \\ \langle 1, -a \rangle & \text{if } b = 1 \text{ and } a > c_1 \text{ (short jobs after cheap prediction)}, \\ \langle 3, -a \rangle & \text{if } b = 0 \text{ and } c_1 + c_2 > a > c_1 \text{ (long jobs, expensive prediction)}, \\ \langle 4, r - a \rangle & \text{if } b = 0 \text{ and } a \geq c_1 + c_2 \text{ (long jobs after expensive prediction)}. \end{cases} \tag{2}$$

Note entering jobs have ranked 2 in the first dimension, placing them behind short jobs awaiting or receiving service. Since after predictions short jobs would simply be placed behind other short jobs, it makes sense to deprioritize the cheap predictions. On the other hand, we prioritize long predictions over long jobs to implement SPRPT.

Finally, for jobs with rank 4 in the first dimension, we use $r - a$ as the secondary rank. Technically the predicted remaining service time is $r - (a - c_1 - c_2)$, since the job's age includes service for predictions of time $c_1 + c_2$. As $c_1$ and $c_2$ are fixed, using $r - a$ is equivalent to using $r - (a - c_1 - c_2)$ for ranking, and we use $r - a$ for convenience.

# C  *SkipPredict* Analysis

**Table 3:** Definition and Description of symbols used in the equations

| Symbol | Description |
|---|---|
| $\mathbb{E}[S'_{<T}] = \int_0^\infty x \cdot p_T(x) \cdot f(x)\,dx$ | Expected service times of predicted short jobs |
| $\mathbb{E}[S'_{\geq T,r}] = \int_{y=0}^r \int_{x=0}^\infty (1 - p_T(x)) \cdot x \cdot g(x,y)dxdy$ | Expected service times of predicted long jobs with service time prediction $\leq r$ |
| $\mathbb{E}[S''_{<T}(c_1)] = \int_0^\infty (x + c_1) \cdot p_T(x) \cdot f(x)\,dx$ | Expected service times of predicted short jobs including prediction cost $c_1$ |
| $\rho'_{<T} = \lambda \int_{x=0}^\infty x f(x) p_T(x)\,dx$ | Load due to predicted short jobs |
| $\rho_r^{ext} = \lambda \left( \mathbb{E}[S'_{<T}] + \mathbb{E}[S'_{\geq T,r}] \right)$ | Load due to predicted short jobs, predicted long jobs with service time prediction $\leq r$ |
| $\rho_{PS}^{srv} = \lambda \left( c_1 + \mathbb{E}[S'_{<T}] \right)$ | Load due to predicted short jobs and their cheap prediction cost |
| $\rho_r^{srv} = \lambda \left( \mathbb{E}[S''_{<T}(c_1)] + (c_1 + c_2) \cdot Q(T,r) \right)$ $+ \lambda \mathbb{E}[S''_{\geq T,r}(c_1 + c_2)]$ | Load due to predicted short jobs, predicted long jobs with service time prediction $\leq r$, and prediction jobs |
| $a(r) = \int_{t=r}^\infty \int_{x=t-r}^\infty (1 - p_T(x)) g(x,t)$ $\cdot (x - (t - r))^2\,dx\,dt$ | Integral related to predicted long jobs with service time prediction $> r$ |
| $Q(T,r) = \int_{y=r}^\infty \int_{x=0}^\infty (1 - p_T(x)) \cdot g(x,y)dxdy$ | Probability of a job being predicted as long with service time prediction $> r$ |

## C.1  External cost model

**Lemma 1.** *For* SkipPredict *in the external cost model, the expected mean response time for a predicted short job,* $\mathbb{E}[T]_{ext}^{PS}$ *is*

$$\mathbb{E}[T]_{ext}^{PS} = \frac{\lambda \mathbb{E}[S'^2_{<T}]}{2(1 - \rho'_{<T})} + \mathbb{E}[S'_{<T}].$$

*Proof.* While SOAP can be used to analyze the mean response time for predicted short jobs, this case is straightforward and follows the mean response time in the system for FCFS, which is given by Equation 23.15 in (Harchol-Balter, 2013). □

**Lemma 2.** *For* SkipPredict *in the external cost model, if we let* $a(r) = \int_{t=r}^\infty \int_{x=t-r}^\infty (1 - p_T(x))g(x,t)(x - (t - r))^2\,dx\,dt$, *the expected mean response time for a predicted long job of true size* $x_J$ *and predicted size* $r$ *is*

$$\mathbb{E}[T(x_J, r)]_{ext}^{PL} = \frac{\lambda}{2(1 - \rho_r^{ext})^2} \left( \mathbb{E}[S'^2_{<T}] + \mathbb{E}[S'^2_{\geq T,r}] + a(r) \right) + \int_0^{x_J} \frac{1}{1 - \rho_{(r-a)^+}^{ext}}\,da$$

*Where* $\rho_r^{ext} = \lambda \left( \mathbb{E}[S'_{<T}] + \mathbb{E}[S'_{\geq T,r}] \right)$ *is the load due to jobs of predicted short and jobs predicted long but their service time prediction less than* $r$ *and* $(r - a)^+ = max(r - a, 0)$.

*Proof.* To analyze *SkipPredict* for a predicted long job in the external cost model using SOAP, we first find the worst future rank and then calculate $X^{new}[rank_{d_J}^{worst}(a)]$, $X_0^{old}[r_{worst}]$ and $X_i^{old}[r_{worst}]$ for predicted long job. As described in (1), the rank function for predicted long jobs is monotonic (here the job descriptor is $d_J = [x_J, 1, r]$), and every job's rank is strictly decreasing with age, thus

$J$'s worst future rank is its initial rank, here: $rank_{d_J}^{\text{worst}}(a) = \langle 2, r-a \rangle$ and $r_{worst} = rank_{d_J}^{\text{worst}}(0) = \langle 2, r \rangle$.

$X^{\text{new}}[rank_{d_J}^{\text{worst}}(a)]$: Suppose that a new job $K$ of predicted size $r_K$ arrives when $J$ has age $a_J$. $J$'s delay due to $K$ depends on whether $K$ is predicted to be short or long. If $K$ is predicted short then it will preempt $J$ and be scheduled till completion because it has a higher class. Otherwise, if $K$ has a predicted job size less than $J$'s predicted remaining process time $(r - a_J)$, $K$ will always outrank $J$. Thus

$$X_{x_K}^{new}[\langle 2, r-a \rangle] = \begin{cases} x_K & K \text{ is predicted short} \\ x_K \mathbb{1}(r_K < r-a) & K \text{ is predicted long} \end{cases}$$

$$\mathbb{E}[X^{new}[\langle 2, r-a \rangle]] = \int_0^\infty p_T(x) x f(x) dx + \int_0^{r-a} \int_{x=0}^\infty x \cdot (1 - p_T(x)) g(x,y) dx dy$$

$X_0^{\text{old}}[r_{worst}]$: Whether another job $I$ is original or recycled depends on its prediction as short or long, and in the case it is long, it also depends on its predicted size relative to $J$'s prediction. If $I$ is predicted short, then it remains original until its completion. Otherwise, if $I$ is predicted long, $I$ is original only if $r_I \leq r$, because then until its completion its rank never exceeds $r$.

$$X_{0,x_I}^{\text{old}}[\langle 2, r \rangle] = \begin{cases} x_I & \text{if } I \text{ is predicted short} \\ x_I \mathbb{1}(r_I \leq r) & \text{if } I \text{ is predicted long} \end{cases}$$

$$\mathbb{E}[X_0^{\text{old}}[\langle 2, r \rangle]] = \int_0^\infty p_T(x) x f(x) dx + \int_{y=0}^r \int_{x=0}^\infty (1 - p_T(x)) \cdot x \cdot g(x,y) dx dy$$

$$\mathbb{E}[(X_0^{\text{old}}[\langle 2, r \rangle])^2] = \int_0^\infty p_T(x) x^2 f(x) dx + \int_{y=0}^r \int_{x=0}^\infty (1 - p_T(x)) \cdot x^2 \cdot g(x,y) dx dy$$

$X_i^{\text{old}}[r_{worst}]$: If another job $I$ is predicted long and if $r_I > r$, then $I$ starts discarded but becomes recycled when $r_I - a = r$. This starts at age $a = r_I - r$ and continues until its completion, which will be $x_I - a_I = x_I - (r_I - r)$. Thus, for $i \geq 2$, $X_{i,x_I}^{\text{old}}[\langle 2, r \rangle] = 0$. Let $t = r_I$:

$$X_{1,x_I}^{\text{old}}[\langle 2, r \rangle] = \begin{cases} 0 & \text{if } I \text{ is predicted short} \\ x_I - (t-r) & \text{if } I \text{ is predicted long} \end{cases}$$

$$\mathbb{E}[X_1^{\text{old}}[\langle 2, r \rangle]^2] = \int_{t=r}^\infty \int_{x=t-r}^\infty (1 - p_T(x)) \cdot g(x,t) \cdot (x - (t-r))^2 dx dt$$

Applying Theorem 1 leads to the result.

$\square$

## C.2   Server time cost model

To analyze the server cost model, as explained, considering the rank function defined in (2), we first find the worst future rank of $J$, denoted as $rank_{d_J}^{\text{worst}}$, as follows:

$$rank_{d_J}^{\text{worst}}(a) = \begin{cases} \langle 2, -a \rangle & \text{if } J \text{ is predicted short} \\ \langle 4, r-a \rangle & \text{if } J \text{ is predicted long} \end{cases}$$

**Lemma 3.** *For* SkipPredict *policy in the server time cost model, the expected mean response time for a predicted short job,* $\mathbb{E}[T]_{srv}^{PS}$ *is*

$$\mathbb{E}[T]_{srv}^{PS} = \frac{\lambda \cdot \left(c_1^2 + 2c_1\mathbb{E}[S'_{<T}] + \mathbb{E}[{S'_{<T}}^2]\right)}{2(1 - \rho_{PS}^{srv})} + \mathbb{E}[S'_{<T}]$$

*where* $\rho_{PS}^{srv} = \lambda\left(c_1 + \mathbb{E}[S'_{<T}]\right)$ *is the load due to jobs of predicted short and their cheap prediction cost.*

*Proof.* To analyze *SkipPredict* for predicted short jobs, we calculate $X^{\text{new}}[rank_{d_J}^{\text{worst}}(a)]$, $X_0^{\text{old}}[r_{worst}]$ and $X_i^{\text{old}}[r_{worst}]$ for these predicted short jobs in the server cost model, where the job descriptor in this case is $(b, r) = (1, *)$.

$X^{\text{new}}[rank_{d_J}^{\text{worst}}(a)]$: Let's consider a new job $K$ arriving when $J$ is at age $a_J$ (where $a_J \leq \min(r, x_J)$). The worst rank of $J$ depends on whether $J$ is predicted to be short or long. If $J$ is predicted short, then $J$'s worst future rank is its current rank $\langle 2, -a_J \rangle$. Given that $K$'s initial rank is $\langle 2, 0 \rangle$, at least equivalent to $J$'s worst future rank, the delay $J$ experiences due to $K$ is: $X_{x_K}^{new}[\langle 2, -a_J \rangle] = 0$

$X_0^{\text{old}}[r_{worst}]$: Suppose that $J$ witnesses an old job $I$ of initial size $x_I$. The duration for which $I$ remains *original* depends on whether its prediction is short or long. If $I$ is predicted short, it remains original until completion. Alternatively, if $I$ is predicted long, it would remain original until the cheap prediction phase (lasting $c_1$), after which its rank shifts to $\langle 3, 0 \rangle$.

$$X_{0,x_I}^{\text{old}}[\langle 2, 0 \rangle] = \begin{cases} c_1 + x_I & \text{if } I \text{ is predicted short} \\ c_1 & \text{if } I \text{ is predicted long} \end{cases}$$

$$\mathbb{E}[X_0^{\text{old}}[\langle 2, 0 \rangle]] = c_1 + \int_0^\infty p_T(x) x f(x) dx$$

$X_i^{\text{old}}[r_{worst}]$: There are no instances of recycled jobs because either $I$ completes its service (if predicted short) or it is discarded completely (if predicted long), and thus never gets a rank lower than $\langle 2, 0 \rangle$. Thus, for $i \geq 1$,

$$X_{i,x_I}^{\text{old}}[\langle 2, 0 \rangle] = 0$$

Applying Theorem 1 yields the result.

$\square$

**Lemma 4.** *For* SkipPredict *policy in the server time cost model, the expected mean response time for a predicted long job of true size $x_J$ and predicted size $r$ is*

$$\mathbb{E}[T(x_J, r)]_{srv}^{PL} = \frac{\lambda}{2(1 - \rho_r^{srv})^2} \cdot \left( \mathbb{E}[{S''_{<T}}^2(c_1)] + (c_1 + c_2)^2 \cdot Q(T, r) + \mathbb{E}[{S''_{\geq T,r}}^2(c_1 + c_2)] + a(r) \right)$$

$$+ \int_0^{x_J} \frac{1}{1 - \rho_{(r-a)^+}^{srv}} da$$

*Where* $Q(T, r) = \int_{y=r}^\infty \int_{x=0}^\infty (1 - p_T(x)) \cdot g(x, y) dx dy \quad (r - a)^+ = max(r - a, 0)$

$$\rho_r^{srv} = \lambda \left( \mathbb{E}[S''_{<T}(c_1)] + (c_1 + c_2) \cdot Q(T, r) + \mathbb{E}[S''_{\geq T,r}(c_1 + c_2)] \right)$$

*is the load due to jobs of predicted short and jobs predicted long but their service time prediction less than $r$ along with the load of the jobs predictions.*

$$a(r) = \int_{t=r}^\infty \int_{x=t-r}^\infty (1 - p_T(x)) g(x, t)(x - (t - r))^2 \, dx \, dt$$

*Proof.* Now we calculate $X^{\text{new}}[rank_{d_J}^{\text{worst}}(a)]$, $X_0^{\text{old}}[r_{worst}]$ and $X_i^{\text{old}}[r_{worst}]$ for predicted long jobs in the server cost most, here again the J's descriptor is $((b,r) = (0,r))$.

$X^{\text{new}}[rank_{d_J}^{\text{worst}}(a)]$: $J$'s delay due to $K$ also depends on $K$ is predicted to be short or long.

$$X_{x_K}^{new}[\langle 4, r - a_J \rangle] = \begin{cases} c_1 + x_K & K \text{ is predicted short} \\ c_1 + c_2 + x_K \mathbb{1}(r_K < r - a_J) & K \text{ is predicted long} \end{cases}$$

Employing the joint distribution $g(x,y)$, and setting $\mathbb{1}(r_K < r - a_J) = \int_{y=0}^{r-a_J} g(x_K, y)dy$, we can derive $J$'s expected delay due to any random new job:

$$\begin{aligned} \mathbb{E}[X^{new}[\langle 4, r - a_J \rangle]] = & c_1 + \int_0^\infty p_T(x)xf(x)dx + \int_0^\infty (1 - p_T(x))c_2 f(x)dx \\ & + \int_0^{r-a_J} \int_{x=0}^\infty (1 - p_T(x))xg(x,y)dxdy \end{aligned}$$

$X_0^{\text{old}}[r_{worst}]$: Regardless of $I$'s prediction, an old job has higher priority than $J$, therefore $J$ will be delayed for the duration of $I$'s service.

$$X_{0,x_I}^{\text{old}}[\langle 4, r \rangle] = \begin{cases} c_1 + x_I & \text{if } I \text{ is predicted short} \\ c_1 + c_2 + x_I \cdot \mathbb{1}(r_I \leq r) & \text{if } I \text{ is predicted long} \end{cases}$$

$$\begin{aligned} \mathbb{E}[(X_0^{\text{old}}[\langle 4, r \rangle])]] = & \int_0^\infty p_T(x)(c_1 + x)f(x)dx + \int_{y=r}^\infty \int_{x=0}^\infty (1 - p_T(x)) \cdot g(x,y)(c_1 + c_2)\, dxdy \\ & + \int_{y=0}^r \int_{x=0}^\infty (1 - p_T(x)) \cdot g(x,y)(c_1 + c_2 + x)\, dxdy \end{aligned}$$

$$\begin{aligned} \mathbb{E}[(X_0^{\text{old}}[\langle 4, r \rangle])^2]] = & \int_0^\infty p_T(x)(c_1 + x)^2 f(x)dx + \int_{y=r}^\infty \int_{x=0}^\infty (1 - p_T(x)) \cdot g(x,y)(c_1 + c_2)^2\, dxdy \\ & + \int_{y=0}^r \int_{x=0}^\infty (1 - p_T(x)) \cdot g(x,y)(c_1 + c_2 + x)^2\, dxdy \end{aligned}$$

$X_i^{\text{old}}[r_{worst}]$: If $J$ is predicted long, job $I$ may be recycled. This occurs when $I$ is predicted as long and with an expensive prediction $r_I > r$. $I$ is initially considered as *original* and is served during the cheap and expensive prediction phases, then discarded. At age $a_I = r_I - r$, $I$ is recycled and served till completion, which will be $x_I - a_I = x_I - (r_I - r)$. For $i \geq 2$, $X_{i,x_I}^{\text{old}}[\langle 2, r \rangle] = 0$, let $t = r_I$:

$$X_{1,x_I}^{\text{old}}[\langle 4, r \rangle] = \begin{cases} 0 & \text{if } I \text{ is predicted short} \\ x_I - (t - r) & \text{if } I \text{ is predicted long} \end{cases}$$

$$\mathbb{E}[X_1^{\text{old}}[\langle 4, r \rangle]^2] = \int_{t=r}^\infty \int_{x=t-r}^\infty (1 - p_T(x)) \cdot g(x,t) \cdot (x - (t-r))^2 \cdot dxdt$$

Applying Theorem 1 yields the result.

□

**Lemma 5.** *The mean response time for predicted long jobs in the external cost model is*

$$\mathbb{E}[T_{ext}^{PL}] = \frac{\int_{x=0}^\infty \int_{y=0}^\infty (1 - p_T(x))g(x,y)\mathbb{E}[T(x,y)]_{ext}^{PL} dydx}{\int_{x=0}^\infty \int_{y=0}^\infty (1 - p_T(x))g(x,y)dydx}$$

*and in the server cost model is:*

$$\mathbb{E}[T_{srv}^{PL}] = \frac{\int_{x=0}^{\infty}\int_{y=0}^{\infty}(1 - p_T(x))g(x,y)\mathbb{E}[T(x,y)]_{srv}^{PL}dydx}{\int_{x=0}^{\infty}\int_{y=0}^{\infty}(1 - p_T(x))g(x,y)dydx}$$

# D   Baselines Analysis

## D.1   SPRPT Analysis

In SPRPT the job descriptors only include the job size and service time predictions, e.g. $\mathcal{D} = [\text{size}, \text{predicted time}] = [x, r]$. Thus, the rank function in the external cost model is:

$$rank_{ext}([x, r], a) = r - a. \tag{3}$$

In the server cost model:

$$rank_{srv}([x, b], a) = \begin{cases} \langle 1, -a \rangle & \text{if } 0 \leq a \leq c_1 \text{ (initial rank, scheduling prediction)}, \\ \langle 2, r - a \rangle & \text{if } a > c_1 \text{ (jobs after prediction)}. \end{cases} \tag{4}$$

**Lemma 6.** *For SPRPT in the external cost model, the expected mean response time for a job of true size $x_J$ and predicted size $r$ is*

$$\mathbb{E}[T(x_J, r)]_{ext}^{SPRPT} = \frac{\lambda}{2(1 - \rho'_r)^2} \left( \int_{y=0}^{r} \int_{x_I=0}^{\infty} x_I^2 \cdot g(x_I, y) dx_I dy \right.$$

$$\left. + \int_{t=r}^{\infty} \int_{x_I=t-r}^{\infty} g(x_I, t) \cdot (x_I - (t - r))^2 \cdot dx_I dt \right) + \int_0^{x_J} \frac{1}{1 - \rho'_{(r-a)^+}} da,$$

*where $\rho'_r = \lambda \int_{y=0}^{r} \int_{x_I=0}^{\infty} x_I \cdot g(x_I, y) dx_I dy$.*

*Proof.* SPRPT has a rank function $rank(r, a) = r - a$ for job of size $x$ and predicted size $r$. As this rank function is monotonic, $J$'s worst future rank is its initial prediction:

$$rank_{d_J, x_J}^{\text{worst}}(a) = r - a.$$

When $a_J = 0$, the rank function is denoted by $r_{worst} = rank_{d_J, x_J}^{\text{worst}}(0) = r$.

$X^{\text{new}}[rank_{d_J, x_J}^{\text{worst}}(a)]$ **computation:**   Suppose that a new job $K$ of predicted size $r_K$ arrives when $J$ has age $a$. If $K$ has a predicted job size less than $J$'s predicted remaining process time $(r - a)$, $K$ will always outrank $J$. Thus

$$X_{x_K}^{new}[r - a] = x_K \mathbb{1}(r_K < r - a)$$

$$\mathbb{E}[X^{new}[r - a]] = \int_0^{r-a} \int_{x_K=0}^{\infty} x_K \cdot g(x_K, y) dx_K dy$$

$X_0^{\text{old}}[r_{worst}]$ **computation:**   Whether job $I$ is an original or recycled job depends on its predicted size relative to J's predicted size. If $r_I \leq r$, then $I$ is original until its completion because its rank never exceeds $r$.

$$X_{0, x_I}^{\text{old}}[r] = x_I \mathbb{1}(r_I \leq r).$$

$$\mathbb{E}[X_0^{\text{old}}[r]] = \int_{y=0}^{r} \int_{x_I=0}^{\infty} x_I \cdot g(x_I, y) dx_I dy.$$

$$\mathbb{E}[(X_0^{\text{old}}[r])^2]] = \int_{y=0}^{r} \int_{x_I=0}^{\infty} x_I^2 \cdot g(x_I, y) dx_I dy.$$

$X_i^{\text{old}}[r_{worst}]$ **computation:** If $r_I > r$, then $I$ starts discarded but becomes recycled when $r_I - a = r$. This means at age $a = r_I - r$ and served till completion, which will be $x_I - a_I = x_I - (r_I - r)$, let $t = r_I$:

Thus, we have

$$X_{1,x_I}^{\text{old}}[r] = x_I - (t - r).$$

For $i \geq 2$,

$$X_{i,x_I}^{\text{old}}[r] = 0.$$

$$\mathbb{E}[X_1^{\text{old}}[r]^2] = \int_{t=r}^{\infty} \int_{x_I=t-r}^{\infty} g(x_I, t) \cdot (x_I - (t-r))^2 \cdot dx_I dt.$$

Applying Theorem 5.5 of SOAP (Scully and Harchol-Balter, 2018) yields that the mean response time of jobs with descriptor $(r)$ and size $x_J$ is as follows. Let

$$\rho_r' = \lambda \int_{y=0}^{r} \int_{x_I=0}^{\infty} x_I \cdot g(x_I, y) dx_I dy.$$

Then

$$\mathbb{E}[T(x_J, r)]_{ext}^{SPRPT} = \frac{\lambda \left( \int_{y=0}^{r} \int_{x_I=0}^{\infty} x_I^2 \cdot g(x_I, y) dx_I dy + \int_{t=r}^{\infty} \int_{x_I=t-r}^{\infty} g(x_I, t) \cdot (x_I - (t-r))^2 \cdot dx_I dt \right)}{2(1 - \rho_r')^2}$$
$$+ \int_0^{x_J} \frac{1}{1 - \rho_{(r-a)+}'} da.$$

Let $f_p(y) = \int_{x=0}^{\infty} g(x, y) dx$. Then the mean response time for a job with size $x_J$, and the mean response time over all jobs are given by

$$\mathbb{E}[T(x_J)] = \int_{y=0}^{\infty} f_p(y) \mathbb{E}[T(x_J, y)] dy,$$

$$\mathbb{E}[T]_{ext}^{SPRPT} = \int_{x=0}^{\infty} \int_{y=0}^{\infty} g(x, y) \mathbb{E}[T(x, y)] dy dx.$$

$\square$

**Lemma 7.** *For SPRPT in the server time cost model, the expected mean response time for a job of true size $x_J$ and predicted size $r$ is*

$$\mathbb{E}[T(x_J, r)]_{srv}^{SPRPT} = \frac{\lambda}{2(1 - \rho_r'')^2} \left( \int_{y=r}^{\infty} \int_{x=0}^{\infty} c_2^2 \cdot g(x, y) dx dy + \int_{y=0}^{r} \int_{x_I=0}^{\infty} (c_2 + x_I)^2 \cdot g(x_I, y) dx_I dy \right.$$
$$\left. + \int_{t=r}^{\infty} \int_{x_I=t-r}^{\infty} g(x_I, t) \cdot (x_I - (t-r))^2 \cdot dx_I dt \right) + \int_0^{x_J} \frac{1}{1 - \rho_{(r-a)+}''} da,$$

*where $\rho_r'' = \lambda \left( c_2 + \int_{y=0}^{r} \int_{x_I=0}^{\infty} x_I \cdot g(x_I, y) dx_I dy \right)$.*

*Proof.* $X^{\text{new}}[rank_{d_J}^{\text{worst}}(a)]$: $J$'s worst future rank is $\langle 2, r - a_J \rangle$. In this case, $J$'s delay due to a new job $K$ is $c_2$ plus $x_K$ is its predicted service time less than $J$'s remaining process time.

$$X_{x_K}^{new}[\langle 2, r - a_J \rangle] = c_2 + x_K \mathbb{1}(r_K < r - a_J).$$

$$\mathbb{E}[X^{new}[\langle 2, r - a_J \rangle]] = c_2 + \int_0^{r-a} \int_{x_K=0}^{\infty} x_K \cdot g(x_K, y) dx_K dy.$$

$X_0^{\textbf{old}}[r_{worst}]$ **computation:** In this model, old job $I$ delays $J$ at least $c_2$. In addition, If $r_I \leq r$, then $I$ is original until its completion because its rank never exceeds $r$.

$$X_{0,x_I}^{\text{old}}\langle 2, r - a_J\rangle = c_2 + x_I \mathbb{1}(r_I \leq r).$$

$$\mathbb{E}[X_0^{\text{old}}\langle 2, r - a_J\rangle] = c_2 + \int_{y=0}^{r}\int_{x_I=0}^{\infty} x_I \cdot g(x_I, y)dx_I dy.$$

$$\mathbb{E}[(X_0^{\text{old}}\langle 2, r - a_J\rangle)^2]] = \int_{y=r}^{\infty}\int_{x=0}^{\infty} c_2^2 \cdot g(x, y)dxdy + \int_{y=0}^{r}\int_{x_I=0}^{\infty} (c_2 + x_I)^2 \cdot g(x_I, y)dx_I dy.$$

$X_i^{\textbf{old}}[r_{worst}]$ **computation:** If $r_I > r$, then $I$ starts discarded but becomes recycled when $r_I - a = r$. This means at age $a = r_I - r$ and served till completion, which will be $x_I - a_I = x_I - (r_I - r)$, let $t = r_I$:

Thus, we have

$$X_{1,x_I}^{\text{old}}\langle 2, r - a_J\rangle = x_I - (t - r).$$

For $i \geq 2$,

$$X_{i,x_I}^{\text{old}}\langle 2, r - a_J\rangle = 0,$$

$$\mathbb{E}[X_1^{\text{old}}\langle 2, r - a_J\rangle^2] = \int_{t=r}^{\infty}\int_{x_I=t-r}^{\infty} g(x_I, t) \cdot (x_I - (t - r))^2 \cdot dx_I dt.$$

Applying Theorem 5.5 of SOAP (Scully and Harchol-Balter, 2018) yields that the mean response time of jobs with descriptor $(r)$ and size $x_J$ is as follows. Let

$$\rho_r'' = \lambda\left(c_2 + \int_{y=0}^{r}\int_{x_I=0}^{\infty} x_I \cdot g(x_I, y)dx_I dy\right).$$

Then

$$\mathbb{E}[T(x_J, r)]_{srv}^{SPRPT} = \frac{\lambda}{2(1 - \rho_r'')^2}\left(\int_{y=r}^{\infty}\int_{x=0}^{\infty} c_2^2 \cdot g(x, y)dxdy + \int_{y=0}^{r}\int_{x_I=0}^{\infty} (c_2 + x_I)^2 \cdot g(x_I, y)dx_I dy\right.$$

$$\left. + \int_{t=r}^{\infty}\int_{x_I=t-r}^{\infty} g(x_I, t) \cdot (x_I - (t - r))^2 \cdot dx_I dt\right) + \int_0^{x_J} \frac{1}{1 - \rho_{(r-a)+}''}\, da$$

Let $f_p(y) = \int_{x=0}^{\infty} g(x, y)dx$. The mean response time for a job with size $x_J$ and the mean time for a general job are give by

$$\mathbb{E}[T(x_J)] = \int_{y=0}^{\infty} f_p(y)\mathbb{E}[T(x_J, y)]dy,$$

$$\mathbb{E}[T]_{ext}^{SPRPT} = \int_{x=0}^{\infty}\int_{y=0}^{\infty} g(x, y)\mathbb{E}[T(x, y)]dydx.$$

$\square$

## D.2 1bit Analysis

For the 1bit policy, we define the rank function of this approach and then analyze it using SOAP framework. Here the job descriptors only include the job sizes and binary predictions, e.g. $\mathcal{D} = [\text{size, predicted short/long}] = [x, b]$.

$$rank_{ext}([x, b], a) = \begin{cases} \langle 1, -a \rangle & \text{if } b = 1, \\ \langle 2, -a \rangle & \text{if } b = 0. \end{cases} \tag{5}$$

In the server cost model, this approach results in the following rank function:

$$rank_{srv}([x, b], a) = \begin{cases} \langle 2, -a \rangle & \text{if } 0 \leq a \leq c_1 \text{ (initial rank, and cheap prediction)}, \\ \langle 1, -a \rangle & \text{if } b = 1 \text{ and } a > c_1 \text{ (short jobs after cheap prediction)}, \\ \langle 3, -a \rangle & \text{if } b = 0 \text{ and } a > c_1 \text{ (long jobs after cheap prediction)}. \end{cases} \tag{6}$$

The mean response time for predicted short jobs of this approach is similar to predicted short jobs of *SkipPredict* as nothing has changed.

**Lemma 8.** *For the 1bit policy in the external cost model, the expected mean response time for a predicted long job of true size $x_J$ is*

$$\mathbb{E}[T(x_J)]_{ext}^{1bit, PL} = \frac{\lambda}{2(1 - \rho)(1 - \rho_{new}^{ext})} \mathbb{E}[S^2] + \int_0^{x_J} \frac{1}{1 - \rho_{new}^{ext}} \, da,$$

*where $\rho_{new}^{ext} = \lambda \int_0^\infty p_T(x) x f(x) dx$, the load due to predicted short jobs.*

*Proof.* To analyze 1-bit advice for predicted long job in the external cost model using SOAP, we first find the worst future rank and then calculate $X^{new}[rank_{d_J}^{worst}(a)]$, $X_0^{old}[r_{worst}]$ and $X_i^{old}[r_{worst}]$ for predicted long job. For predicted long jobs, the rank function is monotonic as described in (5), therefore J's worst future rank is its initial rank.

$$rank_{d_J}^{worst}(a) = \langle 2, -a \rangle,$$

and $r_{worst} = rank_{d_J}^{worst}(0) = \langle 2, 0 \rangle$.

$X^{new}[rank_{d_J}^{worst}(a)]$: Suppose that a new job $K$ arrives when $J$ has age $a_J$. $J$'s delay due to $K$ depends on $K$ is predicted to be short or long.

Only if $K$ is predicted short then it will preempt $J$ and be scheduled till completion because it has a higher class as long jobs are scheduled also according to FCFS so in case of $K$ predicted long it will not preempt $J$

$$X_{x_K}^{new}[\langle 2, -a \rangle] = \{ x_K \quad K \text{ is predicted short.}$$

$$\mathbb{E}[X^{new}[\langle 2, -a \rangle]] = \int_0^\infty p_T(x) x f(x) dx.$$

$X_0^{old}[r_{worst}]$: Now if old job $I$ either predicted short or long is an original job then it remains original until its completion (regardless of if it is predicted long or short). Thus, for $i \geq 1$, $X_{i, x_I}^{old}[\langle 2, 0 \rangle] = 0$.

$$X_{0, x_I}^{old}[\langle 2, 0 \rangle] = x_I.$$

$$\mathbb{E}[X_0^{old}[\langle 2, 0 \rangle]] = \int_0^\infty x f(x) dx.$$

$$\mathbb{E}[(X_0^{old}[\langle 2, 0 \rangle])^2]] = \int_0^\infty x^2 f(x) dx.$$

Applying Theorem 5.5 of SOAP (Scully and Harchol-Balter, 2018) yields the result:

$$\mathbb{E}[T(x_J)]_{ext}^{PL} = \frac{\lambda}{2(1-\rho)(1-\rho_{new}^{ext})} \int_0^\infty x^2 f(x)\, dx + \int_0^{x_J} \frac{1}{1-\rho_{new}^{ext}}\, da,$$

where $\rho_{new}^{ext} = \lambda \left( \int_0^\infty p_T(x) x f(x) dx \right)$.

As a second way of thinking about this proof, it can be said that this is the original FCFS system with slowdowns caused by subsystem 1 (predicted short jobs) which is $\frac{1}{1-\rho_{new}^{ext}}$.

$\square$

**Lemma 9.** *For the 1bit policy in the server time cost model, the expected mean response time for a predicted long job of true size $x_J$ is*

$$\mathbb{E}[T(x_J)]_{srv}^{1bit,PL} = \frac{\lambda}{2(1-\rho_{c_1})(1-\rho_{new}^{srv})} \int_0^\infty (x+c_1)^2 f(x)\, dx + \int_0^{x_J} \frac{1}{1-\rho_{new}^{srv}}\, da,$$

*where $\rho_{new}^{srv} = \lambda \left( c_1 + \int_0^\infty p_T(x) x f(x) dx \right)$ and $\rho_{c_1} = \lambda \left( \int_0^\infty (x+c_1) f(x) dx \right)$.*

*Proof.* In this case, according to (6), J's worst future rank is

$$rank_{d_J}^{\text{worst}}(a) = \langle 3, -a \rangle$$

and $r_{worst} = rank_{d_J}^{\text{worst}}(0) = \langle 3, 0 \rangle$.

$X^{\text{new}}[rank_{d_J}^{\text{worst}}(a)]$: Let's say $J$ has age $a_J$ when $K$ arrives. The delay caused by $K$ for $J$ depends on whether it is predicted to be short or long for $K$. If $K$ is predicted short then it will preempt $J$ and be scheduled along with its cheap prediction till completion. Otherwise, if $K$ is predicted long, it will delay $J$ only for the cheap prediction.

$$X_{x_K}^{new}[\langle 3, -a_J \rangle] = \begin{cases} c_1 + x_K & K \text{ is predicted short,} \\ c_1 & K \text{ is predicted long.} \end{cases}$$

$$\mathbb{E}[X^{new}[\langle 3, -a \rangle]] = c_1 + \int_0^\infty p_T(x) x f(x) dx.$$

$X_0^{\text{old}}[r_{worst}]$: Each old job is scheduled for cheap prediction (which costs $c_1$), and an old job $I$, regardless of whether it is predicted long or short remains original until its completion. Thus, for $i \geq 1$, $X_{i,x_I}^{\text{old}}[\langle 3, 0 \rangle] = 0$.

$$X_{0,x_I}^{\text{old}}[\langle 3, 0 \rangle] = c_1 + x_I$$

$$\mathbb{E}[X_0^{\text{old}}[\langle 3, 0 \rangle]] = \int_0^\infty (c_1 + x) f(x) dx$$

$$\mathbb{E}[(X_0^{\text{old}}[\langle 3, 0 \rangle])^2]] = \int_0^\infty (c_1 + x)^2 f(x) dx$$

Using Theorem 5.5 of SOAP (Scully and Harchol-Balter, 2018) yields the result:

$$\mathbb{E}[T(x_J)]_{srv}^{1bit,PL} = \frac{\lambda}{2(1-\rho_{c_1})(1-\rho_{new}^{srv})} \int_0^\infty (x+c_1)^2 f(x)\, dx + \int_0^{x_J} \frac{1}{1-\rho_{new}^{srv}}\, da,$$

where $\rho_{new}^{srv} = \lambda \left( c_1 + \int_0^\infty p_T(x) x f(x) dx \right) \quad \rho_{c_1} = \lambda \left( \int_0^\infty (x+c_1) f(x) dx \right)$.

$\square$

# E  *DelayPredict* Analysis

Rank function of *DelayPredict*: We model the system using $\mathcal{D} = [\text{size, predicted time}] = [x, r]$. Here we assume that the service time prediction $r$ is greater than $L$, because jobs that require expensive prediction are longer than $L$. Since a job's age is $L$ when it is preempted and obtains a prediction, in the external model, the predicted remaining time for long jobs is $r - L - (a - L) = r - a$. For the service time model their age after being predicted starts at $L + c_2$, so the predicted remaining time is $r - L - (a - L - c_2) = r - a - c_2$. As $c_2$ is fixed among all jobs, instead of the predicted remaining time we can use $r - a$ as the rank for convenience. Accordingly, *DelayPredict* in the external cost model has the following rank function:

$$rank_{ext}([x, r], a) = \begin{cases} \langle 1, -a \rangle & \text{if } 0 \le a < L \\ \langle 2, r - a \rangle & \text{if } a \ge L \end{cases} \tag{7}$$

In the server cost model, *DelayPredict* results in the following rank function:

$$rank_{srv}([x, r], a) = \begin{cases} \langle 1, -a \rangle & \text{if } 0 \le a < L \text{ (initial rank),} \\ \langle 2, -a \rangle & \text{if } L \le a \le L + c_2 \text{ (expensive prediction calculation),} \\ \langle 3, r - a \rangle & \text{if } a > L + c_2 \text{ (long jobs after expensive prediction).} \end{cases} \tag{8}$$

**Lemma 10.** *For* DelayPredict *in both the external cost model and the server time cost model, the expected mean response time for a short job is*

$$\mathbb{E}[T]_{ext}^{DelayPredict, S} = \mathbb{E}[T]_{srv}^{DelayPredict, S} = \frac{\lambda}{2(1 - \rho_L)} \left( \int_0^L x^2 f(x)\, dx + \int_L^\infty L^2 f(x) dx \right) + \int_0^L x f(x) dx,$$

*where* $\rho_L = \lambda \left( \int_0^L x f(x) dx + \int_L^\infty L f(x) dx \right)$, *the load due to jobs while limiting their sizes to L.*

*Proof.* We first find the worst future rank and then calculate $X^{new}[rank_{d_J}^{worst}(a)]$, $X_0^{old}[r_{worst}]$ and $X_i^{old}[r_{worst}]$ for short jobs. As both the external cost model and the server cost model treat short jobs the same, and their worst future rank is the same, the analysis holds for both.

For short jobs, the rank function is monotonic, therefore J's worst future rank is its initial rank:

$$rank_{d_J}^{worst}(a) = \langle 1, -a \rangle$$

and $r_{worst} = rank_{d_J}^{worst}(0) = \langle 1, 0 \rangle$.

$X^{new}[rank_{d_J}^{worst}(a)]$: Since short jobs have higher priorities than long jobs, and they are scheduled FCFS, a new job does not preempt J. Hence $X_{x_K}^{new}[\langle 1, -a \rangle] = 0$.

$X_0^{old}[r_{worst}]$: If an old job $I$ is short, it remains original until it is completed. Otherwise, it remains original only for $L$ times. Thus, for $i \ge 1$, $X_{i, x_I}^{old}[\langle 1, 0 \rangle] = 0$ and

$$X_{0, x_I}^{old}[\langle 1, 0 \rangle] = \begin{cases} x_I & \text{if } I \text{ is short} \\ L & \text{if } I \text{ is long} \end{cases}$$

$$\mathbb{E}[X_0^{old}[\langle 1, 0 \rangle]] = \int_0^L x f(x) dx + \int_L^\infty L f(x) dx$$

$$\mathbb{E}[(X_0^{old}[\langle 1, 0 \rangle])^2]] = \int_0^L x^2 f(x) dx + \int_L^\infty L^2 f(x) dx$$

Applying Theorem 5.5 of SOAP (Scully and Harchol-Balter, 2018) yields the result

$$\mathbb{E}[T]_{ext}^{DelayPredict, S} = \frac{\lambda}{2(1 - \rho_L)} \left( \int_0^L x^2 f(x)\, dx + \int_L^\infty L^2 f(x) dx \right) + \int_0^L x f(x) dx,$$

where $\rho_L = \lambda \left( \int_0^L x f(x) dx + \int_L^\infty L f(x) dx \right)$.  $\square$

**Lemma 11.** *For* DelayPredict *in the external cost model, the expected mean response time for a long job of true size $x_J$ and predicted size $r$ is*

$$
\mathbb{E}[T(x_J, r)]_{ext}^{\text{DelayPredict}, L} = \frac{\lambda}{2(1 - \rho_{L,r}^{ext})^2} \left( \int_{x=0}^{L} x^2 f(x) dx \right.
$$

$$
+ \int_{y=0}^{r} \int_{x=L}^{\infty} x^2 \cdot g(x,y) dx dy + \int_{y=r}^{\infty} \int_{x=L}^{\infty} L^2 \cdot g(x,y) dx dy
$$

$$
+ \left. \int_{t=r}^{\infty} \int_{x=L+t-r}^{\infty} g(x,t) \cdot (x - L - (t-r))^2 \cdot dx dt \right) + \int_{0}^{x_J} \frac{1}{1 - \rho_{L,(r-a)^+}^{ext}} da,
$$

*where $\rho_{L,r}^{ext} = \lambda \left( \int_{x=0}^{L} x f(x) dx + \int_{y=0}^{r} \int_{x=L}^{\infty} x \cdot g(x,y) dx dy + \int_{y=r}^{\infty} \int_{x=L}^{\infty} L \cdot g(x,y) dx dy \right)$ is the load due to short jobs, long jobs predicted to be less than $r$, and other long jobs with their size limited at L. Here $(r-a)^+ = max(r-a, 0)$.*

*Proof.* To analyze *DelayPredict* for a long job in the external cost model using SOAP, we first find the worst future rank and then calculate $X^{new}[rank_{d_J}^{\text{worst}}(a)]$, $X_0^{old}[r_{worst}]$ and $X_i^{old}[r_{worst}]$ for long job. As described in (7), the rank function for long jobs is monotonic, and every job's rank is strictly decreasing with age, thus $J$'s worst future rank is its initial rank, here: $rank_{d_J}^{\text{worst}}(a) = \langle 2, r-a \rangle$ and $r_{worst} = rank_{d_J}^{\text{worst}}(0) = \langle 2, r \rangle$.

$X^{new}[rank_{d_J}^{\text{worst}}(a)]$: Suppose that a new job $K$ of predicted size $r_K$ arrives when $J$ has age $a_J$. $J$'s delay due to $K$ depends on whether $K$ is short or long. If $K$ is short then it will preempt $J$, since it has a higher priority, and be scheduled till completion. If $K$ is long and has a predicted job size less than $J$'s predicted remaining process time $(r - a_J)$, $K$ will preempt $J$ and proceed until completion. Otherwise, If $K$ is long and has a predicted job size more than $J$'s predicted remaining process time, it will preempt $J$ but will be scheduled only for $L$ time.

Thus

$$
X_{x_K}^{new}[\langle 2, r-a \rangle] = \begin{cases} x_K & K \text{ is short} \\ x_K \mathbb{1}(r_K < r-a) + L \cdot \mathbb{1}(r_K \geq r-a) & K \text{ is long} \end{cases}
$$

$$
\mathbb{E}[X^{new}[\langle 2, r-a \rangle]] = \int_{x=0}^{L} x f(x) dx + \int_{y=0}^{r-a} \int_{x=L}^{\infty} x \cdot g(x,y) dx dy + \int_{y=r-a}^{\infty} \int_{x=L}^{\infty} L \cdot g(x,y) dx dy
$$

$X_0^{old}[r_{worst}]$: Whether another job $I$ is original or recycled depends on whether it is short or long, and in the case it is long, it also depends on its predicted size relative to J's prediction. If $I$ is short, then it remains original until its completion. Alternatively, if $I$ is long, it is original until completion if $r_I \leq r$, otherwise, it is original until $L$.

$$
X_{0,x_I}^{old}[\langle 2, r \rangle] = \begin{cases} x_K & K \text{ is short} \\ x_K \mathbb{1}(r_K < r) + L \cdot \mathbb{1}(r_K \geq r) & K \text{ is long} \end{cases}
$$

$$
\mathbb{E}[X_0^{old}[\langle 2, r \rangle]] = \int_{x=0}^{L} x f(x) dx + \int_{y=0}^{r} \int_{x=L}^{\infty} x \cdot g(x,y) dx dy + \int_{y=r}^{\infty} \int_{x=L}^{\infty} L \cdot g(x,y) dx dy
$$

$$
\mathbb{E}[(X_0^{old}[\langle 2, r \rangle])^2]] = \int_{x=0}^{L} x^2 f(x) dx + \int_{y=0}^{r} \int_{x=L}^{\infty} x^2 \cdot g(x,y) dx dy + \int_{y=r}^{\infty} \int_{x=L}^{\infty} L^2 \cdot g(x,y) dx dy
$$

$X_i^{old}[r_{worst}]$: If another job $I$ is long and if $r_I > r$, then $I$ starts discarded but becomes recycled when $r_I - a = r$. This starts at age $a = r_I - r$ and continues until completion, which will be $x_I - L - a_I = x_I - L - (r_I - r)$. Thus, for $i \geq 2$, $X_{i,x_I}^{old}[\langle 2, r \rangle] = 0$. Let $t = r_I$:

$$X_{1,x_I}^{\text{old}}[\langle 2, r \rangle] = \begin{cases} 0 & \text{if } I \text{ is short} \\ x_I - L - (t - r) & \text{if } I \text{ is long} \end{cases}$$

$$\mathbb{E}[X_1^{\text{old}}[\langle 2, r \rangle]^2] = \int_{t=r}^{\infty} \int_{x=L+t-r}^{\infty} g(x,t) \cdot (x - L - (t - r))^2 \cdot dx dt$$

Applying Theorem 1 leads to the result.

$$\mathbb{E}[T(x_J, r)]_{ext}^{DelayPredict, L} = \frac{\lambda}{2(1 - \rho_{L,r}^{ext})^2} \left( \int_{x=0}^{L} x^2 f(x) dx \right.$$

$$+ \int_{y=0}^{r} \int_{x=L}^{\infty} x^2 \cdot g(x,y) dx dy + \int_{y=r}^{\infty} \int_{x=L}^{\infty} L^2 \cdot g(x,y) dx dy$$

$$\left. + \int_{t=r}^{\infty} \int_{x=L+t-r}^{\infty} g(x,t) \cdot (x - L - (t - r))^2 \cdot dx dt \right) + \int_{0}^{x_J} \frac{1}{1 - \rho_{L,(r-a)^+}^{ext}} da$$

Where $\rho_{L,r}^{ext} = \lambda \left( \int_{x=0}^{L} x f(x) dx + \int_{y=0}^{r} \int_{x=L}^{\infty} x \cdot g(x,y) dx dy + \int_{y=r}^{\infty} \int_{x=L}^{\infty} L \cdot g(x,y) dx dy \right)$.

$\square$

**Lemma 12.** *For* DelayPredict *policy in the server cost model, the expected mean response time for a long job of true size $x_J$ and predicted size $r$ is*

$$\mathbb{E}[T(x_J, r)]_{srv}^{DelayPredict, L} = \frac{\lambda}{2(1 - \rho_{L,r}^{srv})^2} \left( \int_{x=0}^{L} x^2 f(x) dx + \int_{y=0}^{r} \int_{x=L}^{\infty} (x + c_2)^2 \cdot g(x,y) dx dy \right.$$

$$+ \int_{y=r}^{\infty} \int_{x=L}^{\infty} (L + c_2)^2 \cdot g(x,y) dx dy$$

$$\left. + \int_{t=r}^{\infty} \int_{x=L+t-r}^{\infty} g(x,t) \cdot (x - L - (t - r))^2 \cdot dx dt \right) + \int_{0}^{x_J} \frac{1}{1 - \rho_{L,(r-a)^+}^{srv}} da,$$

*where* $\rho_{L,r}^{srv} = \lambda \left( \int_{x=0}^{L} x f(x) dx + \int_{y=0}^{r} \int_{x=L}^{\infty} (x + c_2) \cdot g(x,y) dx dy + \int_{y=r}^{\infty} \int_{x=L}^{\infty} (L + c_2) \cdot g(x,y) dx dy \right)$
*is the load due to short jobs, predictions for long jobs, long jobs predicted to be less than $r$, and other long jobs with their size limited at $L$. Here $(r - a)^+ = max(r - a, 0)$.*

*Proof.* In this case, $J$'s worst future rank is $rank_{d_J}^{\text{worst}}(a) = \langle 3, r - a \rangle$ and $r_{worst} = rank_{d_J}^{\text{worst}}(0) = \langle 3, r \rangle$. Now we calculate $X^{\text{new}}[rank_{d_J}^{\text{worst}}(a)]$, $X_0^{\text{old}}[r_{worst}]$ and $X_i^{\text{old}}[r_{worst}]$ for long jobs in the server cost most.

$X^{\text{new}}[rank_{d_J}^{\text{worst}}(a)]$: $J$'s delay due to $K$ also depends on $K$ is short or long. If $I$ is short, then it remains it is scheduled until its completion. Alternatively, if $I$ is long its prediction is scheduled for $c_2$. In addition, if $r_I \leq r$ then it is scheduled until completion, otherwise it is scheduled until $L$.

$$X_{x_K}^{new}[\langle 3, r - a_J \rangle] = \begin{cases} x_K & K \text{ is short} \\ (c_2 + x_K) \cdot \mathbb{1}(r_K < r - a_J) + (c_2 + L) \cdot \mathbb{1}(r_K \geq r - a_J) & K \text{ is long} \end{cases}$$

$$\mathbb{E}[X^{new}[\langle 3, r - a \rangle]] = \int_{x=0}^{L} x f(x) dx + \int_{y=0}^{r-a} \int_{x=L}^{\infty} (x + c_2) \cdot g(x,y) dx dy$$

$$+ \int_{y=r-a}^{\infty} \int_{x=L}^{\infty} (L + c_2) \cdot g(x,y) dx dy$$

$X_0^{\text{old}}[r_{worst}]$: The analysis is similar to the new arrival job. Whether another job $I$ is original or recycled depends on whether it is short or long, and in the case it is long, it also depends on its predicted size relative to J's prediction.

$$X_{0,x_I}^{\text{old}}[\langle 3, r \rangle] = \begin{cases} x_K & K \text{ is short} \\ (c_2 + x_K) \cdot \mathbb{1}(r_K < r) + (c_2 + L) \cdot \mathbb{1}(r_K \geq r) & K \text{ is long} \end{cases}$$

$$\mathbb{E}[X_0^{\text{old}}[\langle 3, r \rangle]] = \int_{x=0}^{L} x f(x)dx + \int_{y=0}^{r} \int_{x=L}^{\infty} (x + c_2) \cdot g(x,y)dxdy + \int_{y=r}^{\infty} \int_{x=L}^{\infty} (L + c_2) \cdot g(x,y)dxdy$$

$$\mathbb{E}[(X_0^{\text{old}}[\langle 3, r \rangle])^2]] = \int_{x=0}^{L} x^2 f(x)dx + \int_{y=0}^{r} \int_{x=L}^{\infty} (x + c_2)^2 \cdot g(x,y)dxdy$$
$$+ \int_{y=r}^{\infty} \int_{x=L}^{\infty} (L + c_2)^2 \cdot g(x,y)dxdy$$

$X_i^{\text{old}}[r_{worst}]$: As described before, if another job $I$ is long and if $r_I > r$, then $I$ starts discarded but becomes recycled when $r_I - a = r$. This starts at age $a = r_I - r$ and continues until completion, which will be $x_I - L - a_I = x_I - L - (r_I - r)$. Thus, for $i \geq 2$, $X_{i,x_I}^{\text{old}}[\langle 3, r \rangle] = 0$. Let $t = r_I$:

$$X_{1,x_I}^{\text{old}}[\langle 3, r \rangle] = \begin{cases} 0 & \text{if } I \text{ is short} \\ x_I - L - (t - r) & \text{if } I \text{ is long} \end{cases}$$

$$\mathbb{E}[X_1^{\text{old}}[\langle 3, r \rangle]^2] = \int_{t=r}^{\infty} \int_{x=L+t-r}^{\infty} g(x,t) \cdot (x - L - (t - r))^2 \cdot dxdt$$

Applying Theorem 1 leads to the result.

$$\mathbb{E}[T(x_J, r)]_{srv}^{DelayPredict,L} = \frac{\lambda}{2(1 - \rho_{L,r}^{srv})^2} \left( \int_{x=0}^{L} x^2 f(x)dx + \int_{y=0}^{r} \int_{x=L}^{\infty} (x + c_2)^2 \cdot g(x,y)dxdy \right.$$
$$+ \int_{y=r}^{\infty} \int_{x=L}^{\infty} (L + c_2)^2 \cdot g(x,y)dxdy$$
$$\left. + \int_{t=r}^{\infty} \int_{x=L+t-r}^{\infty} g(x,t) \cdot (x - L - (t - r))^2 \cdot dxdt \right) + \int_{0}^{x_J} \frac{1}{1 - \rho_{L,(r-a)^+}^{srv}} da$$

Where $\rho_{L,r}^{srv} = \lambda \left( \int_{x=0}^{L} x f(x)dx + \int_{y=0}^{r} \int_{x=L}^{\infty} (x + c_2) \cdot g(x,y)dxdy + \int_{y=r}^{\infty} \int_{x=L}^{\infty} (L + c_2) \cdot g(x,y)dxdy \right)$.

$\square$

# F  Simulation

## F.1  Datasets descriptions

### F.1.1  Real-world datatsets

For real-world traces, we use datasets from Amvrosiadis et al. (Amvrosiadis et al., 2018).

**TwoSigma Dataset:** The workload traces originate from two data centers of TwoSigma, a hedge fund company. The workload comprises data analytics jobs processing financial information. A portion of these jobs utilize Spark (Salloum et al., 2016), while the remaining jobs are handled by proprietary data analytics frameworks developed in-house. The dataset spans a period of 9 months, starting in January 2016, across the two data centers' operations, encompassing a total of 1313 identical compute nodes equipped with 31512 CPU cores and 328TB of RAM. The logs contains $265,029$ jobs and were collected by an internally-developed job scheduler running on top of Mesos (Hindman et al., 2011).

**Google Dataset:** In 2012, Google released a trace of jobs that ran on one of their compute clusters. The workload encompasses both long-running services and batch jobs (Verma et al., 2015). Some of these were issued through the MapReduce framework, and executed on 12583 heterogeneous nodes in May 2011. The dataset we used contains $385,072$ jobs. Google has not disclosed the exact hardware specifications of each cluster node in this trace.

**Trinity Dataset:** Trinity is the largest supercomputer at Los Alamos National Laboratory (LANL), dedicated to capability computing. Capability clusters are large-scale, high-demand resources introducing cutting-edge hardware technologies that aid in achieving significant computing milestones, such as higher-resolution climate and astrophysics models. Trinity's hardware was deployed in two pre-production phases before being fully operational, and the trace was collected before the completion of the second phase. Trinity consisted of 9408 identical compute nodes, totaling 301056 Intel Xeon E5-2698v3 2.3GHz cores and 1.2PB of RAM, making it the largest cluster with a publicly available trace by number of CPU cores. The dataset we used contains $18,872$ jobs collected by the MOAB scheduler.

Their system leverages machine learning to predict job runtimes in large clusters, utilizing features like user IDs, job names, and input sizes. For each job, we have the actual runtime and predicted runtime in seconds, with runtimes summed across tasks for multi-task jobs. We only consider successfully completed jobs and normalize runtimes to a mean of 1, consistent with the synthetic traces.

### Service time predictor for the real-world datasets

For these datasets, (Amvrosiadis et al., 2018) used JVuPredict, the job service time predictor of the JamaisVu scheduling system (Tumanov et al., 2016).

JVuPredict is a runtime (service time) prediction module that is part of the JamaisVu system. Its primary objective is to predict the runtime of a job when it is submitted, utilizing historical data regarding past job characteristics and runtimes. JVuPredict deviates from conventional approaches by attempting to identify jobs that repeat, even when successive runs are not explicitly declared as repeats. This approach proves to be more effective because only the relevant historical data pertaining to the newly submitted job is utilized to generate the runtime estimate. To achieve this, JVuPredict leverages various features of submitted jobs, such as user identifiers and job names, to construct multiple independent predictors. These predictors are then evaluated based on the accuracy they achieve on historical data, and the most accurate predictor is selected for generating future predictions. Once a prediction is made for a new job, that job's data is added to the historical dataset, and the accuracy scores of each predictor model are recalculated. Based on the updated accuracy scores, a new predictor is chosen, and the process is repeated for subsequent job submissions. In general, JVuPredict employs a dynamic approach, continuously refining its prediction models by incorporating new job data and selecting the most accurate predictor for each new job submission.

### F.1.2  Synthetic datasets

For the synthetic traces, we considered two job service distributions; exponentially distributed with mean 1 ($f(x) = e^{-x}$) and the Weibull distribution with cumulative distribution $F = 1 - e^{-\sqrt{2x}}$. The Weibull distribution is heavy-tailed, so that while the average service time remains 1, there are many

more very long jobs than with the exponential distribution. For predictions, we used two prediction models; exponential and uniform predictions. Table 4 summarizes the quantities $p_T(x)$ and $g(x, y)$ for our prediction models. Each of the two-stage predictors could be from a different model.

### F.1.3 Simulation

In our simulation, each data point is obtained by simulating initially empty queues over $1,000,000$ time units. The average response time is calculated for all jobs that terminate after time $100,000$. This process is repeated over 100 simulations, and the reported data point represents the average response time across these 100 simulations. We implemented the simulation in Python 3.7.6. The evaluation was performed on an AMD EPYC 7313 16-Core Processor running Ubuntu 20.04.6 LTS with Linux kernel $5.4.0 - 172$-generic.

| Model | $p_T(x)$ | $g(x, y)$ |
|---|---|---|
| Perfect Prediction | $1$ if $x < T$ | $e^{-x}$ |
| Exponential Prediction | $1 - e^{-\left(\frac{T}{x}\right)}$ | $e^{\frac{-x-y}{x}}$ |
| Uniform Prediction | $\begin{cases} 0 & \text{if } T \leq (1-\alpha)x, \\ 1 & \text{if } T \geq (1+\alpha)x, \\ \frac{T-(1-\alpha)x}{2\alpha x} & \text{otherwise} \end{cases}$ | $\frac{1}{2\alpha x} e^{-x}$ |

**Table 4:** Prediction Models and their Functions

## F.2 Simulation with real-world datasets

Here we present the cost vs. $T$ simulations of the the rest of the real-world datasets, Twosigma and Google.

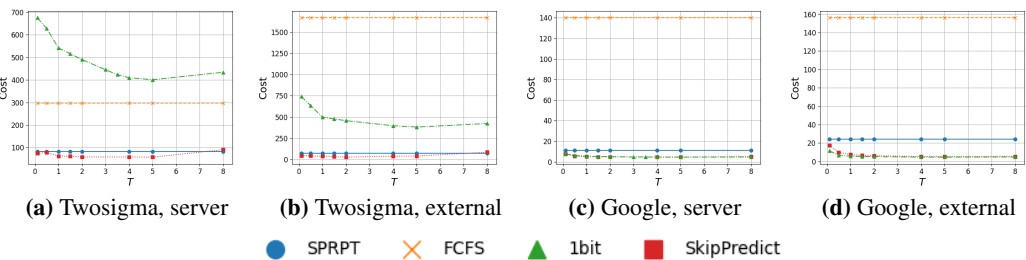

(a) Twosigma, server    (b) Twosigma, external    (c) Google, server    (d) Google, external

● SPRPT    ✕ FCFS    ▲ 1bit    ■ SkipPredict

**Figure 8:** Cost vs. $T$ (with $\lambda = 0.6$) in real-world datasets (Twosigma and Trinity) in both the external and service time models. The default costs for the external model are $c_1 = 0.5, c_2 = 20$, and in the server time are $c_1 = 0.05, c_2 = 0.5$.

## F.3 Simulation with synthetic datasets

Here we present some additional simulations of the the Weibull distribution using both predictors.

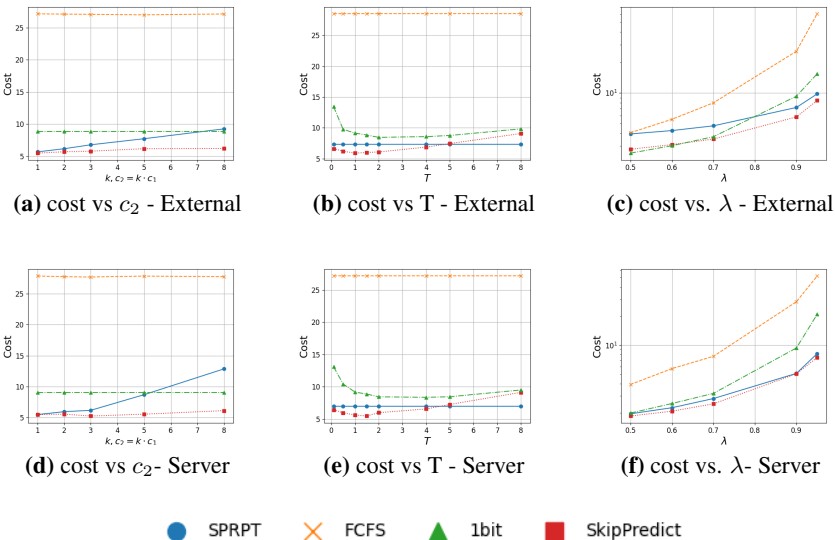

**(a)** cost vs $c_2$ - External  **(b)** cost vs T - External  **(c)** cost vs. $\lambda$ - External

**(d)** cost vs $c_2$- Server  **(e)** cost vs T - Server  **(f)** cost vs. $\lambda$- Server

● SPRPT  ✕ FCFS  ▲ 1bit  ■ SkipPredict

**Figure 9:** Cost in the external cost and server cost models using exponential predictor for Weibull service time. The default costs for the external model are $c_1 = 0.5, c_2 = 2$ and for the server cost model are $c_1 = 0.01, c_2 = 0.05$ (a + d) Cost vs. $c_2$ when $\lambda = 0.9$ and $T = 1$ (b + e) Cost vs. $T$ when $\lambda = 0.9$ (c + f) Cost vs. $\lambda$ when $T = 1$.

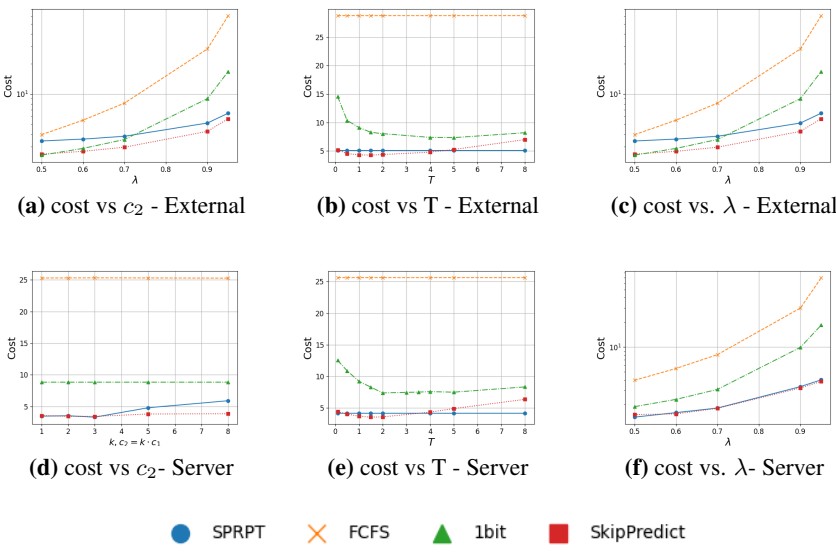

**(a)** cost vs $c_2$ - External  **(b)** cost vs T - External  **(c)** cost vs. $\lambda$ - External

**(d)** cost vs $c_2$- Server  **(e)** cost vs T - Server  **(f)** cost vs. $\lambda$- Server

● SPRPT  ✕ FCFS  ▲ 1bit  ■ SkipPredict

**Figure 10:** Cost in the external cost and server cost models using the uniform predictor for Weibull service time. The default costs for the external model are $c_1 = 0.5, c_2 = 2$ and for the server cost model are $c_1 = 0.01, c_2 = 0.05$ (a + d) Cost vs. $c_2$ when $\lambda = 0.9$ and $T = 1$ (b + e) Cost vs. $T$ when $\lambda = 0.9$ (c + f) Cost vs. $\lambda$ when $T = 1$.

