# OpenReview forum: "SkipPredict: When to Invest in Predictions for Scheduling"
_NeurIPS.cc/2024/Conference — NeurIPS 2024 poster_

### Official Review · Reviewer_kmdF · 2024-07-08

**Soundness:** 3
**Presentation:** 3
**Contribution:** 2
**Rating:** 5
**Confidence:** 3

**Summary:**

The paper studies the effectiveness of machine-learned predictions for queuing systems, and falls into the area of learning-augmented algorithms / algorithms with predictions.
In particular it studies the M/G/1 queue with poisson arrival and i.i.d. service times with the objective to minimize the average response time.
This problem has previously studies with service time predictions or with single-bit predictions, indicating whether a job is 'short' or 'long' depending on a certain threshold. Moreover, previously it has been assumed that predictions are for free.
In contrast to these previous works, in this paper an algorithm needs to 'pay' for receiving predictions. This payment can either be some additional cost (external cost), or an additional load to the machine (server time cost) which delays the execution of other jobs. The latter is motivated by the scenario that the prediction itself first needs to be computed. Moreover, in this paper an algorithm has access to both, (cheaper) 1bit predictions and (more expensive) service time predictions.

The authors propose the algorithm 'SkipPredict' which first queries a 1bit prediction for every job that arrives, and in case that this returns 'long', also queries the service time prediction. 'Short' jobs are generally preferred and scheduled First-Come-First-Serve (FCFS). Furthermore, they study a second algorithm 'DelayPredict' which only uses service time predictions. In contrast to SkipPredict, it starts scheduling every job until it can be categorized as being 'long', and only then queries a service time prediction.

The authors compare both algorithms to the baselines FCFS, 1bit (only cheap predictions) and SRPPT (only expensive predictions). They find that in theory and in empirical experiments, the benefit of SkipPredict and DelayPredict depends on the how the threshold, the system load and the costs for both prediction types relate.

**Strengths:**

- The idea of using a two-stage prediction setup composed of a cheap but less expressive and an expensive but more expressive prediction is to the best of my knowledge new in the area of learning-augmented algorithms, and might have an impact on this field.
- I really like the idea that the generation of a prediction requires processing volume itself. I think such a model could also be interesting for other scheduling problems, also in adversarial models.
- The results show that the use of this two-stage prediction setup help to lower the overall cost compared to the baselines for certain instances.

**Weaknesses:**

- I think the main weakness of this work is that the results seem not to be very surprising. In particular, I think it is clear that the benefit of such a two-stage algorithm like SkipPredict heavily depends on how the costs relate. I am somehow missing some hard statements which summarize the findings more precisely.
- Moreover, the two main components of SkipPredict, 1bit and SRPPT, have been previously studied and analyzed. Thus, to me it seems that the paper does not provide many interesting techniques or analyses.

Given the conceptually interesting models, I tend to rate this paper as borderline right now. However, I am not too familiar with queueing theory and, thus, cannot for sure assess the weight of the second weakness I mentioned.

**Questions:**

Typos:
- Line 114 "somewhat"
- Line 224 missing whitespace

**Limitations:**

-

---

> ### Author Rebuttal · Authors · 2024-08-06
>
> We appreciate your thoughtful feedback and understand your concerns. We believe the perceived "lack of surprise" in our results may stem from our presentation, which we will improve in the revision.
> - While it may seem intuitive that the effectiveness of a two-stage algorithm depends on relative costs, our key contribution is the formalization of this intuition through precise, closed-form formulas of two new algorithms (SkipPredict and DelayPredict). These formulas calculate the mean response time of jobs with size predictions, accounting for prediction costs. This quantitative approach can provide exact thresholds for when predictions become beneficial, moving beyond just high-level qualitative understanding.
>
> - To better illustrate the potential applications of our work, we plan to discuss how our approach could be relevant to practical scenarios to show how our formulas might be used to assess the trade-offs of utilizing predictions.
>
> - We appreciate the opportunity to clarify the point on the use of 1bit and SPRPT (previous works that have been studied before). First, the previous analysis of 1bit and SPRPT did not take costs into account;  this is one of our contributions. Second, SkipPredict (and its analysis) does not directly rely on the 1bit and SPRPT analyses in deriving the closed-form average response time. Rather, SkipPredict represents a novel approach that, in extreme cases, converges to these known algorithms:
> - When the threshold T is set to infinity, SkipPredict converges to the 1bit (or more clearly, 1bit with analysis that takes costs into account).
> - When the threshold T is set to 0, it converges to SPRPT (or more clearly, SPRPT with analysis that takes costs into account).
> For a general T, neither 1bit nor SPRPT can be used as components. Their analyses are included primarily as baselines. and because the previous analyses of 1bit and SPRPT were incomplete and arguably impractical because they did not consider costs. Moreover, our DelayPredict algorithm doesn't make use of cheap prediction (1bit) at all -- instead, we use the idea of having an initial bound on processing act as an implicit "prediction". We will clarify this in the revised manuscript to better convey the innovation and importance of our approach in addressing this significant gap in previous research
>
> - We agree that our paper would benefit from more explicit statements summarizing our key findings. In our revision, we will include a dedicated section that clearly articulates our main results and their implications, emphasizing the novel contributions of our work.
>
> - Thanks for pointing out the typos, we will fix them in the revised paper.

---

> > ### Comment · Reviewer_kmdF · 2024-08-13
> >
> > I thank the authors for their rebuttal. I have the impression that several issues can be indeed fixed by a strongly refined presentation. As I wrote in my initial review, the paper introduces several interesting concepts, which are interesting for the area of learning-augmented algorithms. Thus, I decided to slightly raise my score from 4 to 5.
> > However, I would like to leave it open to the AC/SACs on how to value potential improvements via a strongly refined presentation.

---

### Official Review · Reviewer_oree · 2024-07-12

**Soundness:** 2
**Presentation:** 3
**Contribution:** 2
**Rating:** 4
**Confidence:** 2

**Summary:**

This paper studies a scheduling problem that aims to minimize the expected response time, where jobs arrive online and the algorithm needs to decide the priority of jobs. This paper proposes a new algorithm called skip-prediction. Namely, the algorithm first uses a cheap prediction to partition the whole job set into two parts: long job set and short job set. For jobs in short job set, they have the highest priority and no prediction will be made. And, the algorithm ranks these short jobs by the first-come-first-serve rule. For jobs in long job sets, an expensive prediction will be made to predict the size of these long jobs. Then, the algorithm ranks these long jobs using the shortest remaining processing time first rule.
The paper considers two different models; in the first model, predictions consume some extra costs and in the second model, predictions consume server times. For each model, the total cost of the algorithm is defined as the total expected response time plus the prediction cost. The authors provide theoretical proof to compute the formula for the total cost of the proposed skip-prediction algorithm, and then run an experiment on some datasets.

**Strengths:**

The general motivation of the paper is good. Namely, it considers the case where the prediction does not get for free. The algorithm needs to optimize the scheduling objective with the prediction cost together. On the positive side, I expect that it will have a positive impact on practice application. I also expect that one may be able to abstract some interesting theoretical models from this paper; so it’s likely to influence future work.

**Weaknesses:**

I have to say that I am usually working on the theoretical analysis of the algorithms (e.g., approximation or online algorithms). I am not in the right position to judge the quality and novelty of the experimental paper. To me, the presentation of the paper is not clear; the authors describe an algorithm in the model section instead of defining a problem. Besides this, the authors give the formula of the total cost of skip-prediction in Table 1, but it is not clear how good the quality of these formulas is from the theoretical perspective.

**Questions:**

N.A.

**Limitations:**

N.A.

---

> ### Author Rebuttal · Authors · 2024-08-06
>
> Thank you for recognizing the motivation behind our work and its potential impact on practical applications. We appreciate your concern about the clarity of the presentation, particularly the flow from problem definition to algorithm description. We understand this may have obscured the primary contribution of our paper.
> To recap our objectives and contributions (see also the general Author Rebuttal):
> - Our goal: To develop a scheduling framework that accounts for the cost of predictions, addressing a gap in current learning-augmented algorithms for scheduling.
> - Why this goal: In real data systems, predictions consume resources and time. The common assumption in learning-augmented algorithms that predictions are cost-free does not reflect realistic scenarios.
> - Our main contribution: Comparison and analysis of multiple new algorithms (SkipPredict and DelayPredict). We note we also re-analyzed previous algorithms (1bit and SPRPT) under our more realistic models that take cost into account, extending the previous work. We derive closed-form formulas that calculate the mean response time of jobs with size predictions, given prediction costs. We show that when prediction costs are set to zero, our formulas for our new algorithms align with our new formulas for prior algorithms, demonstrating the robustness and generalizability of this analysis approach. In the appendix, we also explain how to generalize our formulas to non-fixed costs.
>
> These formulas are both theoretically significant and practically applicable. For instance, if a user can estimate prediction costs (using profiling), our formulas can be used to determine the necessary thresholds that when utilizing predictions will yield performance gains.
> We aim to enhance the presentation of our work in the following ways:
> - We recognize that the presentation of the problem definition before the algorithm introduction could be improved. In revising, we will look at restructuring this section to enhance clarity of the problem definition and highlight our end results more clearly.
> - We will provide additional context to better illustrate the significance of the formulas we have derived and their generalizations.

---

### Official Review · Reviewer_ckzA · 2024-07-13

**Soundness:** 4
**Presentation:** 4
**Contribution:** 3
**Rating:** 7
**Confidence:** 4

**Summary:**

Motivated by recent prediction based scheduling of ML jobs in data centers, the paper considers the problem of prediction cost aware scheduling to optimize mean response time. It considers two cost models - external and server time. The paper proposes a novel algorithm, SkipPredict, that uses a two level hierarchical prediction with a cheap short/long prediction at the first level and an expensive total size prediction at the second level. It then uses this information to schedule short jobs as FCFS and long jobs by SRPT. The paper presents an analysis of the expected response time for both models using the SOAP framework and present a comparison against FCFS, one-level FCFS+SRPT and SRPT. Finally, it provides experimental evidence of SkipPredict’s superior performance on synthetic and real world datasets.

**Strengths:**

- The paper is the first to consider the important aspect of cost of prediction in scheduling
- The proposed algorithm, SkipPredict, elegantly utilizes an efficient hierarchical scheme and is practical to implement
- Thorough and well presented analysis of SkipPredict as well as the baselines including generalisation to non-fixed costs
- Baselines compared against are comprehensive

**Weaknesses:**

- While the motivation for the problem comes from scheduling ML jobs, the algorithm and analysis are solely queueing theoretic arguments which makes me question the relevance/interest of this work to the NeurIPS community. MLSys/SIGMETRICS perhaps might be a better fit.
- Further, I believe none of the experiments presented reflect the characteristics of a machine learning job. The Exponential and Weibull distributions of service considered in the synthetic data are unlike the very light tailed (almost deterministic) nature of ML jobs. The real world datasets consist of CPU computations which make them unlikely to be from ML jobs.

**Questions:**

- While the paper analytically shows that SkipPredict does better than the baselines for any general service distribution, it would also be good to see this in the experiments with distributions/datasets representing ML jobs
- (Minor) It would be nice to have a short explanation of how the SOAP framework is used in the main paper as well instead of just in the appendix.

---

> ### Author Rebuttal · Authors · 2024-08-06
>
> We would like to thank the reviewer for the feedback and the thoughtful suggestions.
>
> - We understand the concern about the relevance of our queuing theoretic approach to NeurIPS. Our approach relies on queuing theory to develop algorithms that efficiently manage resources in systems, opening new directions in learning-augmented algorithms, a topic of growing importance in the NeurIPS community. As mentioned by reviewer R-OTX6, there has been related previous work on prediction costs for more abstract problems that have appeared in this community, with more limitations (such as a budget on the number of predictions), rather than our cost optimization framework. Unlike typical approaches in learning-augmented algorithms that assume free predictions, our work challenges this by addressing the realistic scenario of dedicated resources for jobs. This is particularly important in scheduling, where, as in our Server Cost model, the same server uses time to make the prediction as well as to execute the work of the jobs being scheduled. We believe our work will inspire new models and algorithms in the broader context of learning-augmented algorithms. Importantly, these ideas have potential applications in various predictive systems, including those for Large Language Models (LLMs), where efficient resource management and accurate cost assessment of predictions are crucial for optimal performance.
>
> - We appreciate the reviewer's observation regarding the characteristics of ML jobs. To clarify:
>
> (1) Our evaluation framework is designed to capture scheduling for general jobs, with ML-type jobs being a subset of these. Our theoretical results hold for any general distribution, including those typical of ML workloads.
>
> (2) The presented real datasets focus on scheduling jobs with predicted service times in large-scale distributed systems, which aligns with previous work (e.g., Mitzenmacher and Dell’Amico; The Supermarket Model With Known and Predicted Service Times).
>
> (3) We appreciate the reviewer's point about better-representing ML job characteristics. We are not aware of, nor have we found through web searches, definitive information about specific statistical distribution models that accurately describe machine learning (ML) job distributions. However, based on general knowledge of job distributions, exponential and normal distributions could potentially model ML jobs. In our paper, we evaluate our approach using exponential and Weibull distributions, and as part of the rebuttal, we have included the results for an example normal distribution in the provided pdf.  (We can readily include more experiments in the final version.) Regarding the real-world datasets we used, if the reviewer or others can provide or suggest real ML datasets, we would be glad to test our approach on them and include the results in our revised paper.
>
> (4) These additions will demonstrate that SkipPredict's performance extends to scenarios that closely resemble ML workloads, complementing our experimental results that show SkipPredict outperforms baselines for the tested service distributions.
>
> - We appreciate this suggestion of adding SOAP framework in the main paper would enhance reader understanding. We will add a concise overview of how SOAP is applied in our work and an explanation of the rank function of the jobs in each model.

---

> ### Comment · Reviewer_ckzA · 2024-08-07
>
> - Acknowledge the recent interest in cost of predictions for learning augmented algorithms and their role in building infrastructure for ML systems
> - Appreciate the additional experiments with the lighter tailed normal distribution
> - Agree that public datasets with ML job distributions are unavailable (or atleast I am unaware of those as well). I would appreciate if the authors include a remark explaining the characteristics of the real-world dataset in the final version.
> - Given the strengths of the paper in the novel model of the costs, solid theory with closed form expressions and well-presented analysis, I increase my score to 7

---

### Official Review · Reviewer_oTX6 · 2024-07-13

**Soundness:** 4
**Presentation:** 3
**Contribution:** 3
**Rating:** 6
**Confidence:** 4

**Summary:**

The paper considers the job scheduling in the M/G/1 queueing model when the system has access to predictions regarding job lengths. The paper explicitly considers the cost incurred for obtaining the predictions in such a system. Two models are considered - (i) external cost: obtaining predictions incurs a fixed cost, but do not affect the service time of a job, and (ii) server time cost: predictions are obtained via operations that themselves need to be scheduled incurring time that delays jobs.

The authors consider a setting in which there are two kinds of predictions - (i) a cheap 1-bit prediction that only classifies whether the job length is above or below a fixed threshold T, and (ii) an expensive prediction that predicts the actual job length.
The authors propose a natural algorithm, called SkipPredict, that first obtains cheap predictions for all jobs to classify each job as either "short" or "long". All jobs classified as "short" are scheduled via FCFS. The algorithm then obtains expensive predictions for all long jobs which are then scheduled using the shortest predicted remaining time first rule.

The authors analyze this algorithm under both cost models using the SOAP analysis framework and obtains explicit expressions for the mean response times of jobs.

**Strengths:**

- The paper raises an interesting question. Most work on learning augmented algorithms assumes that predictions are available to the algorithm for "free" which is certainly not true in practice. Explicitly modeling prediction costs is a good area for further research. The server time cost model that models the delays due to predictions is especially interesting for scheduling problems.

**Weaknesses:**

- The paper is missing some references to prior work. For example,
(i) "Online algorithms with Costly Predictions" (Drygala et al) considers costly predictions in other learning augmented settings.
(ii) "Parsimonious Learning-Augmented Caching" (Im et al) considers caching with goal of using few predictions.
(iii) many papers on learning augmented scheduling.

**Questions:**

N/A

---

> ### Author Rebuttal · Authors · 2024-08-06
>
> We thank the reviewer for supporting our paper.
> We appreciate your feedback on missing references and we agree that these references are relevant to our study.  In particular, we acknowledge the relevance of the paper 'Online algorithms with Costly Predictions', along with the related work 'Advice Querying under Budget Constraint for Online Algorithms'. We believe these papers (which appeared in AISTATS and NeurIPS, respectively) show the relevance of this area and our work to the NeurIPS community. We note these papers addressed standard online problems, such as the ski rental problem. However, we believe that the queueing setting introduces unique complexities not present in these more traditional problems. Notably, much of the literature in this area focuses on 'budgeted' settings, where the number of predictions is limited, whereas our work seeks to optimize overall costs considering both prediction and operational costs.
>
> In our revision, we will:
> - Incorporate the suggested references and other pertinent works into our related works section.
> - Provide a more comprehensive review of learning-augmented scheduling literature.

---

### Author Rebuttal · Authors · 2024-08-06

We thank the reviewers for their thoughtful feedback. We are encouraged they found the paper raises an interesting question (R-oTX6, R-oree) and is the first to consider the important aspect of the cost of prediction in scheduling (R-ckzA, R-kmdF), which is relevant for learning-augmented algorithms, a growing field in the NeurIPS community.
Reviewers R-oTX6, R-oree, and R-kmdF believe our approach will have a positive impact on practical applications and influence future work, potentially resulting in interesting theoretical models derived from this paper (R-oree, R-kmdF). We are glad they found our proposed algorithm elegant (R-ckzA), with thorough and well-presented analysis of SkipPredict  (R-ckzA), comprehensive evaluation compared to baselines (R-kmdF), and achieved significant improvements. In our paper, we evaluate our approach using exponential and Weibull distributions; as a reviewer (R-ckzA) asked to see more results, we have here provided results for a normal distribution example (see uploaded pdf), where we also see benefits from our algorithms.
One concern, reported mainly by reviewers less familiar with the field (R-oree, R-kmdF), was lack of clarify regarding the primary contribution of the paper. Our key contribution is the comparison and analysis of multiple new algorithms (SkipPredict and DelayPredict). We derive closed-form formulas that calculate the mean response time of jobs with size predictions accounting for the prediction cost in two different models (external cost and server time cost) for each algorithm.
More generally, previous works in scheduling with predictions, as well as other related areas where predictions could be used, often ignore the cost of the prediction in the analysis or design of the system, even if it arises in experiments. In this respect, our contribution is to raise the bar for future work by formally incorporating the costs of predictions. We do this through several additional contributions, including:  providing two general cost models; re-analyzing previous algorithms (1bit and SPRPT) with costs; and examining the idea of using multiple, different-cost predictions to improve performance with SkipPredict.
In addition, we acknowledge the relevance of the paper 'Online algorithms with Costly Predictions" suggested by R-oTX6, along with the related work 'Advice Querying under Budget Constraint for Online Algorithms'. We believe these papers (which appeared in AISTATS and NeurIPS, respectively) show the relevance of this area and our work to the NeurIPS community. We note these papers addressed standard online problems, such as the ski rental problem. However, we believe that the queueing setting introduces unique complexities not present in these more traditional problems. Notably, much of the literature in this area focuses on 'budgeted' settings, where the number of predictions is limited, whereas our work seeks to optimize overall costs considering both prediction and operational costs.
We acknowledge that we could have emphasized and elaborated on the formulas more clearly, as they represent the main theoretical advancement of our work. We address some specific questions below and will incorporate all feedback in the final version.

---

### Decision · Program_Chairs · 2024-09-25

**Decision:**

Accept (poster)

**Comment:**

This paper considers how to incorporate costly predictions into scheduling. This is a highly relevant paper for people working on algorithms with predictions. Most of the reviewers though the model and algorithm will be of interest to the community.  The main caveat the reviews discussed was that the results seem not surprising.  I, like some of the reviewers, believe it is the model that is of interest and sufficient enough that this paper should be accepted.